



# NOₓ emissions, isoprene oxidation pathways, vertical mixing, and implications for surface ozone in the Southeast United States

Katherine R. Travis[1], Daniel J. Jacob[1,2], Jenny A. Fisher[3,4], Patrick S. Kim[2], Eloise A. Marais[1], Lei Zhu[1], Karen Yu[1], Christopher C. Miller[1], Robert M. Yantosca[1], Melissa P. Sulprizio[1], Anne M. Thompson[5], Paul O. Wennberg[6,7], John D. Crounse[6], Jason M. St. Clair[6], Ronald C. Cohen[8], Joshua L. Laughner[8], Jack E. Dibb[9], Samuel R. Hall[10], Kirk Ullmann[10], Glenn M. Wolfe[11,12], Illana B. Pollack[13], Jeff Peischl[14,15], Jonathan A. Neuman[14,15], and Xianliang Zhou[16,17]

[1]Department of Earth and Planetary Sciences and School of Engineering and Applied Sciences, Harvard University, Cambridge, Massachusetts, USA
[2]Earth and Planetary Sciences, Harvard University, Cambridge, MA, USA
[3]Centre for Atmospheric Chemistry, School of Chemistry, University of Wollongong, Wollongong, NSW, Australia
[4]School of Earth and Environmental Sciences, University of Wollongong, Wollongong, NSW, Australia
[5]NASA Goddard Space Flight Center, Greenbelt, Maryland, USA
[6]Division of Geological and Planetary Sciences, California Institute of Technology, Pasadena, CA, USA
[7]Division of Engineering and Applied Science, California Institute of Technology, Pasadena, CA, USA
[8]Department of Chemistry, University of California, Berkeley, CA, USA
[9]Earth System Research Center, University of New Hampshire, Durham, NH, USA
[10]Atmospheric Chemistry Division, National Center for Atmospheric Research, Boulder, CO, USA
[11]Atmospheric Chemistry and Dynamics Laboratory, NASA Goddard Space Flight Center, Greenbelt, MD, USA
[12]Joint Center for Earth Systems Technology, University of Maryland Baltimore County, Baltimore, MD, USA
[13]Atmospheric Science Department, Colorado State University, Fort Collins, Colorado, USA
[14]University of Colorado, Cooperative Institute for Research in Environmental Sciences, Boulder, CO, USA
[15]NOAA, Division of Chemical Science, Earth Systems Research Lab, Boulder, CO USA
[16]Department of Environmental Health and Toxicology, School of Public Health, State University of New York at Albany, Albany, New York, USA
[17]Wadsworth Center, New York State Department of Health, Albany, New York, USA

*Correspondence to*: Katherine R. Travis (ktravis@fas.harvard.edu)

**Abstract.** Ozone pollution in the Southeast US involves complex chemistry driven by emissions of anthropogenic nitrogen oxide radicals (NOₓ ≡ NO + NO₂) and biogenic isoprene. Model estimates of surface ozone concentrations tend to be biased high in the region and this is of concern for designing effective emission control strategies to meet air quality standards. We use detailed chemical observations from the SEAC⁴RS aircraft campaign in August and September 2013, interpreted with the GEOS-Chem chemical transport model (CTM) at 0.25°×0.3125° horizontal resolution, to better understand the factors controlling surface ozone in the Southeast US. We find that the National Emission Inventory (NEI) for NOₓ from the US Environmental Protection Agency (EPA) is too high in the Southeast and nationally by 50%. This is demonstrated by SEAC⁴RS observations of NOₓ and its oxidation products, by surface network observations of nitrate wet deposition fluxes, and by OMI satellite observations of tropospheric NO₂ columns. Upper tropospheric NO₂ from lightning makes a large contribution to the satellite observations that must be accounted for when using these data to estimate surface NOₓ emissions.



Aircraft observations of upper tropospheric $NO_2$ are higher than simulated by GEOS-Chem or expected from $NO$-$NO_2$-$O_3$ photochemical stationary state. $NO_x$ levels in the Southeast US are sufficiently low that only half of isoprene oxidation proceeds by the high-$NO_x$ pathway to produce ozone; this fraction is only moderately sensitive to changes in $NO_x$ emissions because isoprene and $NO_x$ emissions are spatially segregated. GEOS-Chem with reduced $NO_x$ emissions provides an

unbiased simulation of ozone observations from the aircraft and from ozonesondes, and reproduces the observed ozone production efficiency in the boundary layer as derived from a regression of ozone and $NO_x$ oxidation products. However, the model is still biased high by $8\pm13$ ppb relative to observed surface ozone in the Southeast US. Ozonesondes launched during midday hours show a 7 ppb ozone decrease from 1.5 km to 0.2 km altitude, whereas GEOS-Chem has no such gradient because of efficient boundary layer mixing. We conclude that model biases in simulating surface ozone over the Southeast

US may be due to a combination of excessive $NO_x$ emissions and excessive boundary layer vertical mixing.

## 1 Introduction

Ground-level ozone is a harmful air pollutant that causes adverse health and environmental impacts. Ozone is produced in the troposphere when volatile organic compounds (VOCs) and carbon monoxide (CO) are photochemically oxidized in the presence of nitrogen oxide radicals ($NO_x \equiv NO+NO_2$). The mechanism for producing ozone is complicated, involving

hundreds of chemical species interacting with transport on multiple time scales. In October 2015, the US Environmental Protection Agency (EPA) set a new National Ambient Air Quality Standard (NAAQS) for surface ozone as a maximum daily 8-h average (MDA8) of 0.070 ppm not to be exceeded more than three times per year. This is the latest in a succession of gradual tightening of the NAAQS from 0.12 ppm (1-h average) to 0.08 ppm in 1997, and to 0.075 ppm in 2008, responding to accumulating evidence that ozone is detrimental to public health even at low concentrations (EPA, 2013).

Chemical transport models (CTMs) tend to significantly overestimate surface ozone in the southeastern US (Lin et al., 2008; Fiore et al., 2009; Reidmiller et al., 2009; Brown-Steiner et al., 2015; Canty et al., 2015). Here we address this issue by using the GEOS-Chem CTM to simulate aircraft observations of ozone and its precursors over the Southeast US in August-September 2013 from the NASA SEAC[4]RS campaign (Toon et al., 2016) and including additional observations from surface networks.

A number of explanations have been proposed for the ozone model biases in the Southeast US. Fiore et al. (2003) suggested excessive modeled ozone inflow from the Gulf of Mexico. Lin et al. (2008) proposed that the ozone dry deposition velocity could be underestimated. McDonald-Buller et al. (2011) pointed out the potential role of halogen chemistry as a sink of ozone. Isoprene is the principal VOC precursor of ozone in the Southeast US in summer, contingent on the supply of $NO_x$

(Chameides et al., 1992). Fiore et al. (2005) found that a major source of uncertainty is the magnitude of isoprene emissions from vegetation and the loss of $NO_x$ through formation of isoprene nitrates. Horowitz et al. (2007) found a large sensitivity of ozone to the fate of isoprene nitrates and the extent to which they release $NO_x$ when oxidized. Squire et al. (2015) found



that the choice of isoprene oxidation mechanism can alter both the sign and magnitude of the response of ozone to isoprene and NO$_x$ emissions.

The SEAC$^4$RS aircraft campaign in August-September 2013 provided an outstanding opportunity to improve our
understanding of ozone chemistry over the Southeast US. The SEAC$^4$RS DC-8 aircraft hosted an unprecedented chemical payload including isoprene and its oxidation products, NO$_x$ and its oxidation products, and ozone. The flights featured extensive boundary layer mapping of the Southeast as well as vertical profiling to the free troposphere (Toon et al., 2016). We use the GEOS-Chem global CTM with high horizontal resolution over North America (0.25°×0.3125°) to simulate and interpret the SEAC$^4$RS observations. We integrate into our analysis additional Southeast US observations during the summer
of 2013 including the NOMADSS aircraft campaign, the SOAS surface site in Alabama, the SEACIONS ozonesonde network, other surface networks, and NO$_2$ satellite data from the OMI instrument. Several companion papers apply GEOS-Chem to simulate other aspects of SEAC$^4$RS and concurrent data for the Southeast US including aerosol sources and optical depth (Kim et al., 2015), isoprene organic aerosol (Marais et al., 2015), organic nitrates (Fisher et al., 2016), formaldehyde and its relation to satellite observations (Zhu et al., 2016), and sensitivity to model resolution (Yu et al., 2016).

**2 GEOS-Chem Model Description**

We use the GEOS-Chem global 3-D CTM (Bey et al., 2001) in version 9.02 (www.geos-chem.org) with modifications described below. GEOS-Chem is driven with assimilated meteorological data from the Goddard Earth Observing System – Forward Processing (GEOS-5.11.0) of the NASA Global Modeling and Assimilation Office (GMAO). The GEOS-5.11.0 data have a native horizontal resolution of 0.25° latitude by 0.3125° longitude and a temporal resolution of 3 h (1 h for
surface variables and mixing depths). We use a nested version of GEOS-Chem (Chen et al., 2009) with native horizontal resolution over North America and adjacent oceans (130° - 60°W, 9.75° - 60°N) and boundary conditions from a global simulation with 4° × 5° horizontal resolution. Boundary layer mixing follows the non-local parameterization implemented by Lin and McElroy (2010). Daytime mixing depths are reduced by 40% as described by Kim et al. (2015) and Zhu et al. (2016) to match lidar observations from SEAC$^4$RS. We conducted the GEOS-Chem nested model simulation for August-September
2013, following six months of initialization at 4° × 5° resolution.

**2.1 Chemistry**

The general features of the chemical mechanism in GEOS-Chem version 9.02 are presented by Mao et al., (2010, 2013). Here, we have aerosol reactive uptake of HO$_2$ produce H$_2$O$_2$, instead of H$_2$O as in Mao et al. (2013), to better match H$_2$O$_2$ observations in SEAC$^4$RS. Isoprene chemistry is of particular interest for our application and we include a number of
updates, listed comprehensively as Supplementary Material (Tables S1 and S2). Companion papers describe the updates relevant to isoprene nitrates (Fisher et al., 2016) and organic aerosol formation (Marais et al., 2015). We focus below on





updates to the low-NO$_x$ pathways for isoprene oxidation where recent lab and field studies have made significant progress. Oxidation of biogenic monoterpenes was also added to the GEOS-Chem mechanism (Fisher et al., 2016) but does not significantly affect ozone.

A critical issue in isoprene chemistry is the fate of the isoprene peroxy radicals (ISOPO$_2$) produced from the oxidation of isoprene by OH (the dominant isoprene sink). When NO$_x$ is sufficiently high, ISOPO$_2$ reacts mainly with NO to produce ozone (high-NO$_x$ pathway). At lower NO$_x$ levels, ISOPO$_2$ may instead react with HO$_2$ or other organic peroxy radicals, or isomerize, in which case ozone is not produced (low-NO$_x$ pathways). Here we increase the molar yield of isoprene hydroperoxide (ISOPOOH) from the ISOPO$_2$ + HO$_2$ reaction to 93.7% using high precision observations of the minor
channels of this reaction from Liu et al. (2013). Oxidation of ISOPOOH by OH produces isoprene epoxides (IEPOX) that subsequently react with OH or are taken up by aerosol (Paulot et al., 2009b; Marais et al., 2015). We use updated rates and products from Bates et al. (2014) for the reaction of IEPOX with OH. We revise the oxidation products of first-generation isoprene nitrates (ISOPN) with OH according to Jacobs et al. (2014).

ISOPO$_2$ isomerization produces hydroperoxyaldehydes (HPALDs) (Peeters et al., 2009; Crounse et al., 2011; Wolfe et al., 2012), and this is now explicitly included in the mechanism. HPALDs go on to react with OH or photolyze at roughly equal rates over the Southeast US. We use the HPALD+OH reaction rate constant from Wolfe et al. (2012) and the products of the reaction from Squire et al. (2015). The HPALD photolysis rate is calculated using the absorption cross-section of MACR, with a quantum yield of 1, as recommended by Peeters and Muller (2010). The photolysis products are taken from Stavrakou
et al. (2010). We include a faster rate constant and revise the product yields for the self-reaction of ISOPO$_2$ according to Xie et al. (2013).

A number of studies have suggested that conversion of NO$_2$ to nitrous acid (HONO) by gas-phase or aerosol-phase pathways could provide a source of HO$_x$ radicals following HONO photolysis (Li et al., 2014; Zhou et al., 2014). This mechanism
would also provide a catalytic sink for ozone when NO$_2$ is produced by the NO + ozone reaction, viz.,

$$NO + O_3 \rightarrow NO_2 + O_2 \tag{1}$$
$$NO_2 \rightarrow HONO \text{ (by various pathways)} \tag{2}$$

$$HONO + h\upsilon \rightarrow NO + OH \tag{3}$$

Observations of HONO from the NOMADSS campaign (https://www2.acom.ucar.edu/campaigns/nomadss) indicate a mean
daytime HONO concentration of 10 ppt in the Southeast US boundary layer (Zhou et al., 2014), whereas the standard gas-phase mechanism in GEOS-Chem version 9.02 yields less than 1 ppt. We added to the mechanism the pathway proposed by Li et al. (2014), in which HONO is produced by the reaction of the HO$_2$•H$_2$O complex with NO$_2$, and reduced the corresponding rate constant to $k_{HO2•H2O+NO2}$ = 2x10$^{-12}$ cm$^3$ molecule$^{-1}$ s$^{-1}$ in order to obtain ~10 ppt daytime HONO in the



Southeast US boundary layer. The resulting impact on boundary layer ozone concentrations is negligible. HONO production may also take place by photolysis of aerosol nitrate (Ye et al., 2016).

## 2.2 Dry Deposition

The GEOS-Chem dry deposition scheme uses a resistance-in-series model based on Wesely (1989) as implemented by Wang

et al. (1998). Underestimate of dry deposition has been invoked as a cause for model overestimates of ozone in the eastern US (Lin et al., 2008; Walker, 2014). Daytime ozone deposition is determined principally by stomatal uptake. Here, we decrease the stomatal resistance from 200 s m$^{-1}$ for both coniferous and deciduous forests (Wesely, 1989) by 20% to match summertime measurements of the ozone dry deposition velocity for a pine forest in North Carolina (Finkelstein et al., 2000) and for the Ozarks oak forest in southeast Missouri (Wolfe et al., 2015), both averaging 0.8 cm s$^{-1}$ in the daytime. The mean

ozone deposition velocity in GEOS-Chem along the SEAC$^4$RS boundary layer flight tracks in the Southeast US averages 0.7±0.3 cm s$^{-1}$ for the daytime (9-16 local) surface layer. Deposition is suppressed in the model at night due to both stomatal closure and near-surface stratification, consistent with the Finkelstein et al. (2000) observations.

Deposition flux measurements for isoprene oxidation products at the Alabama SOAS site (http://soas2013.rutgers.edu)

indicated higher deposition velocities than simulated by GEOS-Chem for isoprene oxidation products (Nguyen et al., 2015). As an expedient, Nguyen et al. (2015) scaled the Henry's law coefficients to enable the deposition velocities calculated by GEOS-Chem to match those measured at the Alabama SOAS site. We follow their approach here. Other important depositing species include HNO$_3$ and peroxyacetyl nitrate (PAN), with mean deposition velocities along the SEAC$^4$RS Southeast US flight tracks in daytime of 3.7 cm s$^{-1}$ and 0.7 cm s$^{-1}$, respectively.

## 2.3 Emissions

We use hourly US anthropogenic emissions from the 2011 EPA national emissions inventory (NEI11v1) at a horizontal resolution of 0.1° × 0.1° and adjusted to 2013 using national annual scaling factors (EPA, 2015). The total national NO$_x$ emission in NEI11v1 for 2013 is 3.5 Tg N. Initial implementation of this inventory in GEOS-Chem resulted in a 60% overestimate of SEAC$^4$RS DC-8 observations for NO$_x$ and HNO$_3$, and a 71% overestimate of nitrate (NO$_3^-$) wet deposition

fluxes measured by the National Acid Deposition Program (NADP) across the Southeast US. This suggests that NEI11v1 NO$_x$ emissions are biased high. Errors in NO$_x$ sources from soils, wildfire, or lightning cannot account for the overestimate because their magnitudes are small relative to fuel combustion, as shown below.

Emissions from power plant stacks, which represent 12% of the NEI11v1 NO$_x$ emissions on an annual basis (EPA, 2015),

are well constrained by continuous emission monitors. Other components of the NEI inventory are more uncertain. A number of studies have found that NEI emission estimates for mobile sources may be too high by a factor of two or more





(Fujita et al., 2012; Brioude et al., 2013; Anderson et al., 2014). Lu et al. (2015) find good agreement between NEI emissions and top-down estimates from OMI $NO_2$, but they assume an error on NEI emissions of 50%.

Here we reduce NEI11v1 emissions (adjusted to 2013) by 60% for all sources except power plants, or 53% of the annual NEI11v1 emissions. The resulting US anthropogenic $NO_x$ emissions for 2013 total 1.7 Tg N $a^{-1}$. As shown in the next section, this reduction largely corrects the bias in the simulation of observations for $NO_x$ and its oxidation products. Soil $NO_x$ emissions, including emissions from fertilizer application, are computed according to Hudman et al. (2012), with a 50% reduction in the Midwest US based on a previous comparison with OMI $NO_2$ observations (Vinken et al., 2014). Open fire emissions are from the daily Quick Fire Emissions Database (QFED) (Darmenov and da Silva, 2014) with diurnal variability from the Western Regional Air Partnership (Air Sciences, 2005). We emit 40% of open fire $NO_x$ emissions as PAN and 20% as $HNO_3$ to account for fast oxidation taking place in the fresh plume (Alvarado et al., 2010). Following Fischer et al. (2014), we inject 35% of fire emissions above the boundary layer, evenly between 3.5 and 5.5 km altitude.

We constrain the lightning $NO_x$ source with satellite data as described by Murray et al. (2012). Lightning $NO_x$ is mainly released at the top of convective updrafts following Ott et al. (2010). During SEAC$^4$RS, we treat the lightning $NO_x$ yield in the Southeast as tropical (250 mol/flash) rather than midlatitudes (500 mol/flash) to be consistent with the tropical nature of convection during SEAC$^4$RS (Toon et al., 2016), and to achieve unbiased upper tropospheric $NO_x$ and ozone profiles.

Figure 1 gives the resulting surface $NO_x$ emissions for the Southeast US for August and September 2013. With the original NEI inventory, fuel combustion accounted for 81% of total surface $NO_x$ emissions in the Southeast US (not including lightning). After reducing NEI emissions, the contribution from fuel combustion is still 68%.

Biogenic VOC emissions are from MEGAN v2.1, including isoprene, acetone, acetaldehyde, monoterpenes, and >$C_2$ alkenes. We reduce MEGAN v2.1 isoprene emissions by 15% to better match SEAC$^4$RS observations (Wolfe et al., 2015; Zhu et al., 2016). Yu et al. (2016) show the resulting isoprene emissions for the SEAC$^4$RS period.

## 3 Constraints on $NO_x$ emissions from aircraft and observations

Figure 2 shows simulated and observed median vertical distributions of $NO_x$, total inorganic nitrate (gas-phase $HNO_3$+aerosol $NO_3^-$), and ozone concentrations along the SEAC$^4$RS flight tracks over the Southeast US. Here and elsewhere the data exclude urban plumes as diagnosed by $[NO_2] > 8$ ppb, open fire plumes as diagnosed by $[CH_3CN] > 200$ ppt, and stratospheric air as diagnosed by $[O_3]/[CO] > 1.25$ mol mol$^{-1}$. These filters exclude <1%, 6%, and 6% of the data respectively. We would not expect the model to be able to capture these features even at native resolution (Yu et al., 2016).





Model results in Figure 2 are shown both with the original $NO_x$ emissions (dashed line) and with non-power plant NEI combustion emissions decreased by 60% (solid line). Decreasing emissions corrects the model bias for $NO_x$ and also largely corrects the bias for inorganic nitrate. Boundary layer ozone is overestimated by 12 ppb with the original $NO_x$ emissions but this bias disappears after decreasing the $NO_x$ emissions.

Further support for decreasing $NO_x$ emissions is offered by observed nitrate wet deposition fluxes from the NADP network (NADP, 2007). Figure 3 compares simulated and observed fluxes for the model with decreased $NO_x$ emissions. Model values have been corrected for precipitation bias following the method of Paulot et al. (2014), in which the monthly deposition flux is assumed to scale to the $0.6^{th}$ power of the precipitation bias. We diagnose precipitation bias in the GEOS-

FP data relative to high-resolution PRISM observations (http://prism.oregonstate.edu). For the Southeast US, the precipitation bias is -34% in August and -21% in September 2013.

We see from Figure 3 that the model with decreased $NO_x$ emissions reproduces the spatial variability in the observations with minimal bias across the US. In comparison, the model with original emissions had a 60% overestimate of the nitrate wet

deposition flux nationally and a 71% overestimate in the Southeast. Thus the need to decrease $NO_x$ emissions relative to NEI applies to the whole US, not just the Southeast. The high deposition fluxes along the Gulf of Mexico, both in the model and in the observations, reflect particularly large precipitation.

The model with decreased $NO_x$ emissions also well reproduces the spatial distribution of $NO_x$ in the Southeast US boundary

layer as observed in SEAC⁴RS. This is shown in Figure 4 with simulated and observed concentrations of $NO_x$ along the flight tracks below 1.5 km altitude. The spatial correlation coefficient is 0.7. There is no indication of regional patterns of model bias that would point to the need for a more selective adjustment of $NO_x$ emissions. Our simple approach of decreasing non-power plant NEI anthropogenic emissions by 60% performs well.

## 4 Constraints from satellite $NO_2$ data: importance of the upper troposphere

Observations of tropospheric $NO_2$ columns by solar backscatter from the OMI satellite instrument offer an additional constraint on $NO_x$ emissions (Duncan et al., 2014; Lu et al., 2015). We compare the tropospheric columns simulated by GEOS-Chem with the NASA operational retrieval (Level 2, v2.1) (NASA, 2012; Bucsela et al., 2013) and the Berkeley High-Resolution (BEHR) retrieval (Russell et al., 2011). The NASA retrieval has been validated to agree with surface measurements within ± 20% (Lamsal et al., 2014). Both retrievals fit the observed backscattered solar spectra to obtain a

slant tropospheric $NO_2$ column, $\Omega_s$, along the optical path of the backscattered radiation detected by the satellite. The slant column is converted to the vertical column $\Omega_v$ by using an air mass factor (AMF) that depends on the vertical profile of $NO_2$ and on the scattering properties of the surface and the atmosphere (Palmer et al., 2001):



$$\Omega_v = \frac{\Omega_S}{AMF} = \frac{\Omega_S}{AMF_G \int_0^{z_T} w(z)S(z)dz} \qquad (4)$$

In Equation 4, $AMF_G$ is the geometric air mass factor that depends on viewing geometry of the satellite, $w(z)$ is a scattering weight calculated by a radiative transfer model that describes the sensitivity of the backscattered radiation to $NO_2$ as a function of altitude, $S(z)$ is a shape factor describing the normalized vertical profile of $NO_2$ number density, and $z_T$ is the

tropopause. Scattering weights for $NO_2$ retrievals typically increase by a factor of 3 between the surface and the upper troposphere (Martin et al., 2002). Here we use our GEOS-Chem shape factors to re-calculate the AMFs in the NASA and BEHR retrievals as recommended by Lamsal et al. (2014) for comparing model and observations. We filter out cloudy scenes (cloud radiance fraction > 0.5) and bright surfaces (surface reflectivity > 0.3).

Figure 5 shows the mean $NO_2$ tropospheric columns from BEHR, NASA and GEOS-Chem (with $NO_x$ emission reductions applied) over the Southeast US for August-September 2013. The BEHR retrieval is on average 6% higher than the NASA retrieval. GEOS-Chem is on average 11±19% lower than the NASA retrieval and 16±18% lower than the BEHR retrieval. Without decreasing NEI $NO_x$ emissions, GEOS-Chem would be biased high against both retrievals by 26-31%. The low bias in the model with reduced $NO_x$ emissions does not appear to be caused by an overcorrection of emissions but rather the

effect of $NO_x$ in the upper troposphere, as explained below.

The tropospheric $NO_2$ columns over the Southeast US in Figure 5 are not solely determined by boundary layer $NO_x$, but also include a large contribution from the free and upper troposphere. Figure 6 (left panel) shows the mean vertical profile of $NO_2$ number density as measured from the aircraft by two independent instruments (NOAA and UC Berkeley) and simulated by

GEOS-Chem. The observations show a secondary maximum in the upper troposphere above 10 km, absent in GEOS-Chem. It has been suggested that aircraft measurements of $NO_2$ in the upper troposphere could be biased high due to decomposition in the instrument inlet of thermally unstable $NO_x$ reservoirs such as $HNO_4$ and methylperoxynitrate (Browne et al., 2011; Nault et al., 2015; Reed et al., 2015). This could possibly account for the difference between the NOAA and UC Berkeley measurements in the upper troposphere (Nault et al., 2015). At the surface, the median difference is $1.8\times10^9$ molecules $cm^{-3}$

which is within the NOAA and UC Berkeley measurement uncertainties of +/- 0.030 ppbv + 7% and +/- 5%, respectively.

The middle panel of Figure 6 shows the cumulative contributions from different altitudes to the slant $NO_2$ column measured by the satellite, using the median vertical profiles from the left panel and applying mean altitude-dependent scattering weights from the NASA and BEHR retrievals. The boundary layer below 1.5 km contributes only 19-28% of the column.

The upper troposphere above 8 km contributes 32-49% in the aircraft observations and 23% in GEOS-Chem. Much of the observed upper tropospheric $NO_2$ likely originates from lightning and is broadly distributed across the Southeast because of the long lifetime of $NO_x$ at that altitude (Li et al., 2005; Bertram et al., 2007; Hudman et al., 2007). The $NO_2$ vertical profile (shape factor) assumed in the BEHR retrieval does not include any lightning influence, and the Global Modeling Initiative





(GMI) model vertical profile assumed in the NASA retrieval likely underestimates the upper tropospheric $NO_2$ similarly to GEOS-Chem in Figure 6. These underestimates of upper tropospheric $NO_2$ in the retrieval shape factors will cause a negative bias in the AMF and therefore a positive bias in the retrieved vertical columns. This could explain the lower GEOS-Chem column in Figure 5 as compared to the retrievals.

The GEOS-Chem underestimate of observed upper tropospheric $NO_2$ in Figure 6 appears to be driven in part by $NO/NO_2$ partitioning. The right panel of Figure 6 shows the $NO/NO_2$ concentration ratio in GEOS-Chem and in the observations (NOAA for NO, UC Berkeley for $NO_2$). One would expect the $NO/NO_2$ concentration ratio in the daytime upper troposphere to be controlled by photochemical steady-state:

$$NO + O_3 \rightarrow NO_2 + O_2 \tag{5}$$
$$NO + HO_2/RO_2 \rightarrow NO_2 + OH/RO \tag{6}$$
$$NO_2 + h\upsilon \overset{O_2}{\rightarrow} NO + O_3 \tag{7}$$

with reaction (6) playing only a minor role so that $[NO]/[NO_2] \approx k_7/(k_5[O_3])$, defining the $NO$-$NO_2$-$O_3$ photochemical steady state (PSS). The PSS plotted in Figure 6 agrees closely with GEOS-Chem. Such agreement has previously been found when

comparing photochemical models with observed $[NO]/[NO_2]$ ratios from aircraft in the upper troposphere (Schultz et al., 1999) and lower stratosphere (Del Negro et al., 1999). The SEAC[4]RS observations show large departure.

The PSS and GEOS-Chem simulation of the $NO/NO_2$ concentration ratio in Figure 6 use $k_5 = 3.0 \times 10^{-12} \exp[-1500/T]$ $cm^3$ molecule$^{-1}$ s$^{-1}$ and spectroscopic information for $k_7$ from Sander et al. (2011). The $NO_2$ photolysis frequencies $k_7$ computed

locally by GEOS-Chem are on average within 10% of the values determined in SEAC[4]RS from measured actinic fluxes (Shetter and Muller, 1999). It is possible that the strong thermal dependence of $k_5$ has some error, considering that only one direct measurement has been published for the cold temperatures of the upper troposphere (Borders and Birks, 1982). Cohen et al. (2000) found that reducing the activation energy of $k_5$ by 15% improved model agreement in the lower stratosphere. Correcting the discrepancy between simulated and observed $[NO]/[NO_2]$ ratios in the upper troposphere in Figure 6 would

require a similar reduction to the activation energy of $k_5$, but this reduction would negatively impact the surface comparison. This inconsistency of the observed $[NO]/[NO_2]$ ratio with basic theory needs to be resolved, as it affects the inference of $NO_x$ emissions from satellite $NO_2$ column measurements. Notwithstanding this inconsistency, the result remains that $NO_2$ in the upper troposphere makes a significant contribution to the tropospheric $NO_2$ column and hence must be properly accounted for when interpreting the $NO_2$ columns in terms of $NO_x$ emissions.

**5 Isoprene oxidation pathways**

Measurements aboard the SEAC[4]RS aircraft included first-generation isoprene nitrates (ISOPN), isoprene hydroperoxide (ISOPOOH), and hydroperoxyaldehydes (HPALD) (Crounse et al., 2006; Paulot et al., 2009a; St. Clair et al., 2010; Crounse





et al., 2011; Beaver et al., 2012; Nguyen et al., 2015). The measurement uncertainties are large (30%, 40%, and 50%, respectively (Nguyen et al., 2015)). These are unique products of the $ISOPO_2$ + NO, $ISOPO_2$ + $HO_2$, and $ISOPO_2$ isomerization pathways and thus track whether oxidation of isoprene proceeds by the high-$NO_x$ pathway (producing ozone) or the low-$NO_x$ pathways. Figure 2 (bottom row) compares simulated and observed concentrations. All three gases are
restricted to the boundary layer because of their short lifetimes. Mean model concentrations in the lowest altitude bin (Figure 2, approximately 400m above ground) differ from observations by 18% for ISOPN and -50% for HPALD. The GEOS-Chem simulation of organic nitrates including ISOPN is further discussed in Fisher et al. (2016).

The bias for HPALD is within the uncertainty of the kinetics and measurement. Our HPALD source is based on the $ISOPO_2$
isomerization rate constant from Crounse et al. (2011). A theoretical calculation by Peeters et al. (2014) suggests a rate constant that is 1.8× higher, which would reduce the model bias for HPALD and ISOPOOH and increase boundary layer OH by 8%. GEOS-Chem overestimates ISOPOOH by 74% below 1.5 km. Recent work by St. Clair et al. (2015) found that the reaction rate of ISOPOOH + OH to form IEPOX is approximately 10% faster than the rate given by Paulot et al. (2009b), which would further reduce the model overestimate. It is likely that after these changes the GEOS-Chem overestimate of
ISOPOOH would be within measurement uncertainty. For both ISOPOOH and HPALD, GEOS-Chem captures much of the spatial variability (r = 0.8 and 0.7, respectively).

Figure 7 shows the model branching ratios for the fate of the $ISOPO_2$ radical by tracking the mass of $ISOPO_2$ reacting via the high-$NO_x$ pathway ($ISOPO_2$+NO) and the low-$NO_x$ pathways over the Southeast US domain. The branching ratios for the
mixed layer are $ISOPO_2$+NO 54%, $ISOPO_2$+$HO_2$ 26%, $ISOPO_2$ isomerization 15%, and $ISOPO_2$+$RO_2$ 5%. The high-$NO_x$ pathway accounts for less than 50% of the fate of isoprene for large areas of the Southeast US. This reflects in part the spatial segregation of isoprene and $NO_x$ emissions (Yu et al., 2016). This segregation buffers the effect of changing $NO_x$ emissions on the fate of isoprene. Our original simulation with higher total $NO_x$ emissions (unadjusted NEI11v1) had a branching ratio for the $ISOPO_2$+NO reaction of 62%, as compared to 54% in our standard simulation.

**6 Implications for ozone: aircraft and ozonesonde observations**

Figure 2 compares simulated and observed median vertical profiles of ozone concentrations over the Southeast US during SEAC[4]RS. There is no significant bias through the depth of the tropospheric column. The median ozone concentration below 1.5 km is 49 ppb in the observations and 51 ppb in the model. We also find excellent model agreement across the US with the SEACIONS ozonesonde network (Figure 8). The successful simulation of ozone is contingent on the decrease in $NO_x$
emissions relative to the NEI inventory. As shown in Figure 2, a simulation with the unadjusted NEI emissions overestimates boundary layer ozone by 12 ppb.



The model also has some success in reproducing the spatial variability of boundary layer ozone seen from the aircraft, as shown in Figure 4. The correlation coefficient is $r = 0.7$ on the 0.25°×0.3125° model grid, and patterns of high and low ozone concentration are consistent. The highest observed ozone (>75 ppb) was found in air influenced by agricultural burning along the Mississippi River and by outflow from Houston over Louisiana. GEOS-Chem does not capture the extreme values and this probably reflects a dilution effect (Yu et al., 2016).

A critical parameter for understanding ozone production is the ozone production efficiency (OPE) (Liu et al., 1987), defined as the number of ozone molecules produced per molecule of $NO_x$ emitted. This can be estimated from atmospheric observations by the relationship between odd oxygen ($O_x \equiv O_3 + NO_2$) and the sum of products of $NO_x$ oxidation, collectively called $NO_z$ and including inorganic and organic nitrates (Trainer et al., 1993; Zaveri, 2003). The $O_x$ vs. $NO_z$ linear relationship (as derived from a linear regression) provides an upper estimate of the OPE because of rapid deposition of $NO_y$, mainly $HNO_3$ (Trainer et al., 2000; Rickard et al., 2002).

Figure 9 shows the observed and simulated daytime (9-16 local) $O_x$ vs. $NO_z$ relationship in the SEAC[4]RS data below 1.5 km, where $NO_z$ is derived from the observations as $NO_y - NO_x \equiv HNO_3$ + aerosol nitrate + PAN + alkyl nitrates. The resulting OPE from the observations (17.4±0.4) agrees well with GEOS-Chem (16.7±0.3). Previous work during the INTEX-NA aircraft campaign in summer 2004 found an OPE of 8 below 4 km (Mena-Carrasco et al., 2007). By selecting INTEX-NA data only for the Southeast and below 1.5 km we find an OPE of 14.1±1.1 (Figure 9, right panel). The median $NO_z$ was 1.1 ppb during SEAC[4]RS and 1.5 ppb during INTEX-NA, a decrease of approximately 40%. With the original NEI11v1 $NO_x$ emissions (53% higher), the OPE from GEOS-Chem would be 14.7±0.3. Both the INTEX-NA data and the model are consistent with the expectation that OPE increases with decreasing $NO_x$ emissions (Liu et al., 1987).

## 7 Implications for ozone: aircraft and ozonesonde observations

Figure 10 compares maximum daily 8-h average (MDA8) ozone values at the US EPA Clean Air Status and Trends Network (CASTNET) sites in June-August 2013 to the corresponding GEOS-Chem values. The model has a mean positive bias of 8±13 ppb with no significant spatial pattern. The model is unable to match the low tail in the observations, including a significant population with MDA8 ozone less than 20 ppb.

The positive bias in the model for surface ozone is remarkable considering that the model is unbiased relative to aircraft observations below 1.5 km altitude (Figures 2 and 4). A standard explanation for model overestimates of surface ozone over the Southeast US, first proposed by Fiore et al. (2003) and echoed in the review by McDonald-Buller et al. (2011), is excessive ozone over the Gulf of Mexico, which is the prevailing low-altitude inflow. We find that this is not the case. SEAC[4]RS included four flights over the Gulf of Mexico, and Figure 11 compares simulated and observed vertical profiles of



ozone and $NO_x$ concentrations that show no systematic bias. The median ozone concentration in the marine boundary layer is 26 ppb in the observations and 29 ppb in the model. The aircraft observations in Figure 4 also show no indication of a coastal depletion that might be associated with halogen chemistry. Remarkably, the median ozone over the Gulf of Mexico is higher than approximately 8% of MDA8 values at sites in the Southeast.

Median ozone concentrations over the Southeast are 49 ppb in the aircraft observations below 1.5 km (Figure 2) and MDA8 ozone is 40 ppb at CASTNET sites. This indicates a large vertical gradient near the surface that GEOS-Chem does not capture. In GEOS-Chem, air rapidly mixes vertically in the unstable mixed layer, which typically extended to 1.5 km altitude along SEAC[4]RS flight tracks as indicated by aerosol lidar (Zhu et al., 2016). Inspection of the low tail of MDA8 ozone

values below 20 ppb at the CASTNET sites indicates that these are associated with colder-than-average conditions, likely corresponding to low clouds or rain. This supports vertical mixing as a factor responsible for the large gradient between ozone observations at surface sites and from the aircraft.

Further support for a vertical gradient of ozone near the surface, even under midday conditions, is offered by the ozonesonde

observations from the SEACIONS network. Figure 12 shows the median ozonesonde profile over the Southeast US (Huntsville, Alabama and St. Louis, Missouri sites) during SEAC[4]RS as compared to GEOS-Chem below 1.5 km. The model shows minimal gradient below 1.5 km with a small increase in ozone in the lowest model layer. The ozonesondes indicate a decrease of 7 ppb from 1.5 km to 0.2 km (the lowest altitude reported). The GEOS-Chem representation of mixed layer dynamics is fairly standard for CTMs and thus we suggest that excessive boundary layer mixing could contribute to the

model overestimates of ozone in the Southeast US reported by the literature. More work is needed to evaluate and improve the representation of mixed layer dynamics in CTMs.

## 8 Conclusions

We used aircraft (SEAC[4]RS), surface, satellite, and ozonesonde observations from August and September 2013, interpreted with the GEOS-Chem chemical transport model, to better understand the factors controlling surface ozone in the Southeast

US. Models tend to overestimate ozone in that region. Determining the reasons behind this overestimate is critical to the design of efficient emission control strategies to meet the ozone NAAQS.

A major finding from this work is that the EPA National Emission Inventory (NEI11v1) for $NO_x$ (the limiting precursor for ozone formation) is biased high by 50% on the national scale. Evidence for this comes from (1) SEAC[4]RS observations of

$NO_x$ and its oxidation products, (2) NADP network observations of nitrate wet deposition fluxes, and (3) OMI satellite observations of $NO_2$. Presuming no error in emissions from large power plants with continuous emission monitors (12% of unadjusted NEI inventory), we conclude that emissions from other industrial sources and mobile sources must be decreased





by 60% from NEI values. We estimate that anthropogenic $NO_x$ emissions in the US in 2013 were 1.7 Tg N a$^{-1}$, and that fuel combustion still accounts for 68% of surface $NO_x$ emissions in the Southeast US in summer. The rest is mainly from soils.

Our analysis of the OMI $NO_2$ satellite data over the Southeast US reveals that the free troposphere makes a dominant contribution to the $NO_2$ tropospheric column retrieved from the satellite. This reflects in part high upper tropospheric $NO_2$ observed from the aircraft, accounting for 32-49% of the tropospheric column above 8km seen by the satellite. Upper tropospheric $NO_2$ will increase in importance if surface emissions continue to decline in the future. Current retrievals of satellite $NO_2$ data do not properly account for this elevated pool of upper tropospheric $NO_2$, so that the reported tropospheric $NO_2$ columns are biased high. The upper tropospheric $NO_2$ in the aircraft observations requires better understanding because

it is associated with a large departure from conventional $NO$-$NO_2$-$O_3$ photochemical steady state.

Atmospheric oxidation of biogenic isoprene (the dominant VOC precursor of summertime ozone in the Southeast) can proceed by either the high-$NO_x$ pathway (where the isoprene peroxy radical $ISOPO_2$ reacts with NO, producing ozone) or low-$NO_x$ pathways (where $ISOPO_2$ reacts with other peroxy radicals or isomerizes, not producing ozone). Measurements of

isoprene oxidation products from each pathway were made aboard the SEAC$^4$RS aircraft and these are reproduced within measurement and kinetic uncertainty by GEOS-Chem. In GEOS-Chem, only half of isoprene over the Southeast US reacts by the high-$NO_x$ pathway, and this fraction is only weakly sensitive to the magnitude of $NO_x$ emissions because isoprene and $NO_x$ emissions are spatially segregated. Thus it is likely that decreasing $NO_x$ emissions to meet the latest ozone NAAQS will not lead to fundamental changes in the pathways for isoprene chemistry.

Our updated GEOS-Chem simulation with decreased $NO_x$ emissions as described above provides an unbiased simulation of boundary layer and free tropospheric ozone measured from aircraft and ozonesondes during SEAC$^4$RS. Decreasing $NO_x$ emissions is critical to this success as the original model with NEI emissions overestimated boundary layer ozone by 12 ppb. The ozone production efficiency (OPE) inferred from $O_x$ vs. $NO_z$ aircraft correlations in the mixed layer is also well

reproduced. Comparison to the INTEX-NA aircraft observations over the Southeast in summer 2004 indicates a 14% increase in OPE associated with a 40% reduction in $NO_x$ emissions.

Despite the unbiased simulation of boundary layer ozone, GEOS-Chem overestimates MDA8 surface ozone observations in the Southeast US by 8±13 ppb. This appears to be due to excessive vertical mixing. Midday ozonesonde data over the

Southeast US during SEAC$^4$RS indicate a 7 ppb decrease in ozone from 1.5 km to 0.2 km altitude whereas the model shows no such decrease. GEOS-Chem assumes efficient vertical mixing in the unstable mixed layer, as is standard in models, but the observations indicate less efficient mixing. It appears that excessive boundary layer mixing, combined with excessive $NO_x$ emissions, could be responsible for the general bias found in models in simulating surface ozone in the Southeast US.



**Acknowledgements**

We are grateful to the entire NASA SEAC[4]RS team for their help in the field. We thank Tom Ryerson for his measurements of NO and $NO_2$ from the NOAA $NO_yO_3$ instrument. We thank L. Gregory Huey for the use of his CIMS PAN measurements. We thank Fabien Paulot and Jingqiu Mao for their helpful discussions of isoprene chemistry. We thank Christoph Keller for his help in implementing the NEI11v1 emissions into GEOS-Chem. We acknowledge the EPA for providing the 2011 North American emission inventory, and in particular George Pouliot for his help and advice. These emission inventories are intended for research purposes. A technical report describing the 2011-modeling platform can be found at: http://www.epa.gov/ttn/chief/net/2011nei/2011_nei_tsdv1_draft2_june2014.pdf. A description of the 2011 NEI can be found at: http://www.epa.gov/ttnchie1/net/2011inventory.html. This work was supported by the NASA Earth Science Division and by STAR Fellowship Assistance Agreement no. 91761601-0 awarded by the US Environmental Protection Agency (EPA). It has not been formally reviewed by EPA. The views expressed in this publication are solely those of the authors. JAF acknowledges support from a University of Wollongong Vice Chancellor's Postdoctoral Fellowship. This research was undertaken with the assistance of resources provided at the NCI National Facility systems at the Australian National University through the National Computational Merit Allocation Scheme supported by the Australian Government.

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

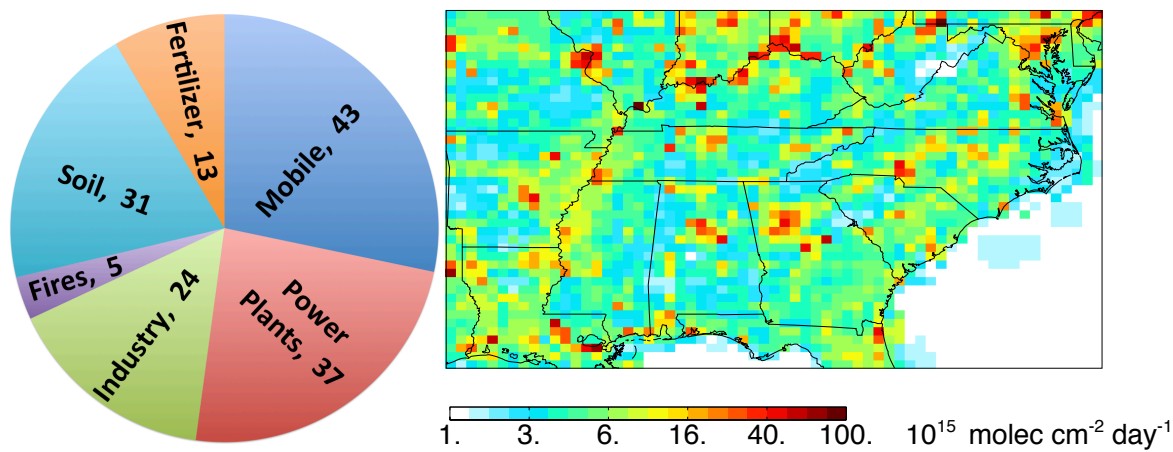

**Figure 1:** Surface $NO_x$ emissions in the Southeast US in GEOS-Chem for August and September 2013 including fuel combustion, soils, fertilizer use, and open fires (total emissions=153 Gg N). Anthropogenic emissions from mobile sources and industry in the National Emission Inventory (NEI11v1) for 2013 have been decreased by 60% to match atmospheric observations (see text). Lightning contributes an additional 25 Gg N to the free troposphere. The emissions are mapped on the 0.25° × 0.3125° GEOS-Chem grid. The pie chart gives the sum of August-September 2013 emissions (Gg N) over the Southeast US domain as shown on the map, and defined as 94.5 to 75° W and 29.5 to 40° N.



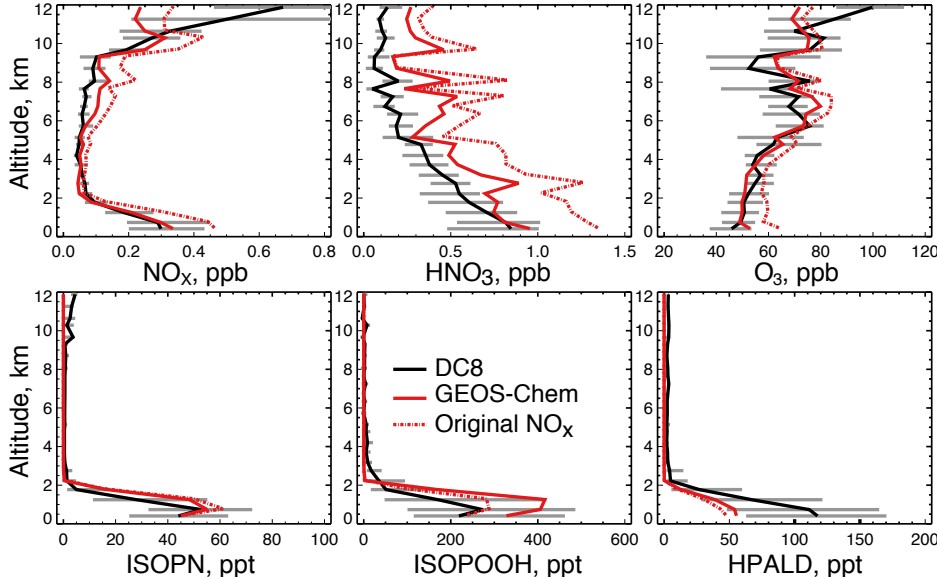

**Figure 2:** Median vertical concentration profiles of $NO_x$, total inorganic nitrate (gas $HNO_3$ + aerosol $NO_3^-$), ozone, isoprene nitrate (ISOPN), isoprene hydroperoxide (ISOPOOH), and hydroperoxyaldehydes (HPALDs) for the SEAC[4]RS flights over the Southeast US (domain of Figure 1). Observations from the DC-8 aircraft are compared to GEOS-Chem model results. The dashed red line shows model results before scaling of $NO_x$ emissions from fuel combustion and lightning. The 25[th] and 75[th] percentiles of the DC-8 observations are shown as grey bars. The SEAC[4]RS observations have been filtered to remove open fire plumes, stratospheric air, and urban plumes as described in the text. Model results are sampled along the flight tracks at the time of flights and gridded to the model resolution. Profiles are binned to the nearest 0.5 km. The NOAA $NO_y O_3$ 4-channel chemiluminescence (CL) instrument made measurements of ozone and $NO_y$ (Ryerson et al., 1998), NO (Ryerson et al., 2000) and $NO_2$ (Pollack et al, 2010). Total inorganic nitrate was measured by the University of New Hampshire Soluble Acidic Gases and Aerosol (UNH SAGA) instrument (Dibb et al., 2003) and was mainly gas-phase $HNO_3$ for the SEAC[4]RS conditions. ISOPOOH, ISOPN, and HPALDs were measured by the Caltech single mass analyzer CIMS (Crounse et al., 2006; Paulot et al., 2009a; Crounse et al., 2011).

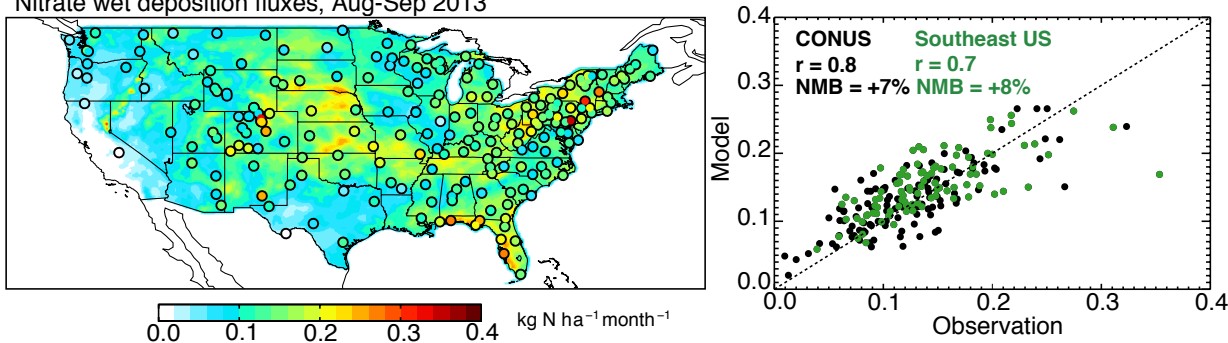

**Figure 3:** Nitrate wet deposition fluxes across the US in August-September 2013. Observations from the NADP network (circles in the left panel) are compared to model values with decreased $NO_x$ emissions (background). Also shown is a scatterplot of simulated versus observed values at individual sites for the whole contiguous US (black) and for the Southeast US (green). The correlation coefficient (r) and normalized mean bias (NMB) are shown inset, along with the 1:1 line.





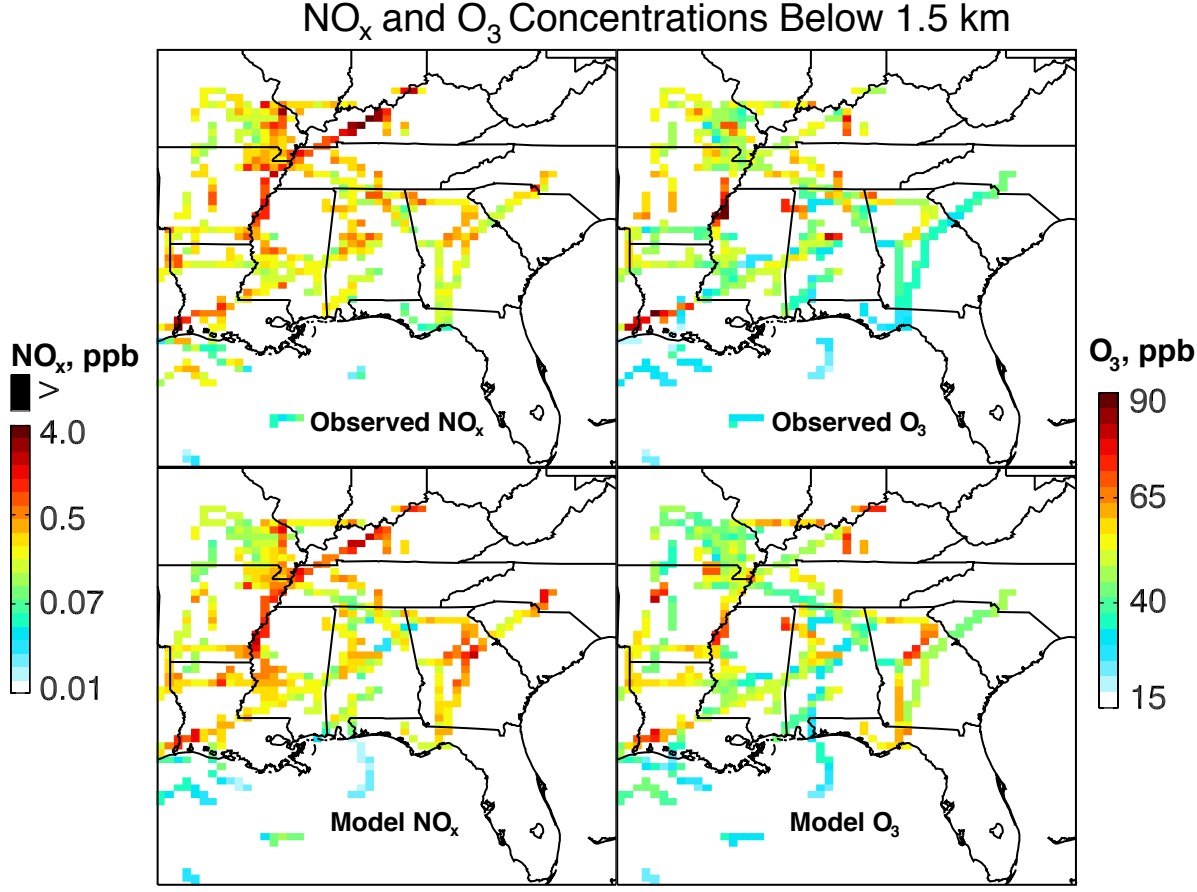

**Figure 4:** Ozone and $NO_x$ concentrations in the boundary layer (0-1.5km) during SEAC[4]RS (6 Aug to 23 Sep 2013). Observations from the aircraft and simulated values are averaged over the $0.25^{\circ} \times 0.3125^{\circ}$ GEOS-Chem grid.



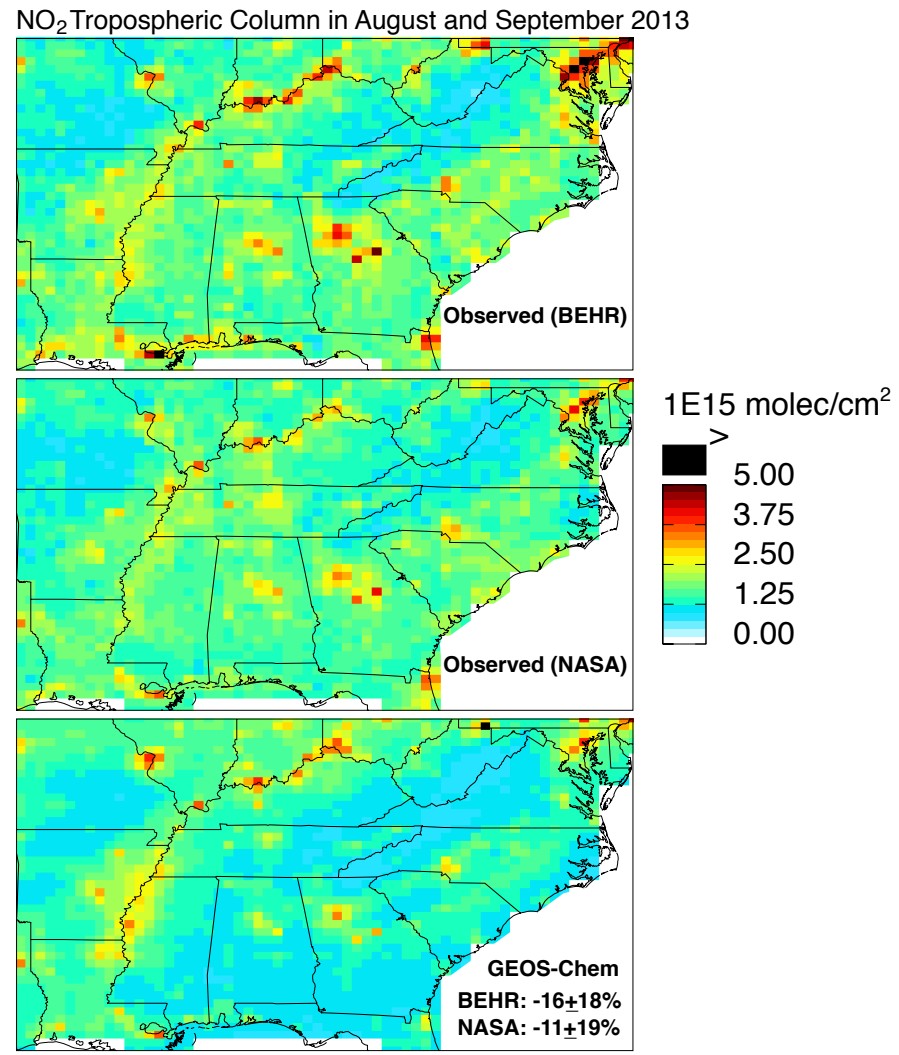

**Figure 5:** NO$_2$ tropospheric columns over the Southeast US in August-September 2013. GEOS-Chem (sampled at the 13:30 local time overpass of OMI) is compared to OMI satellite observations using the BEHR and NASA retrievals. Values are plotted on the 0.25°x0.3125° GEOS-Chem grid. The GEOS-Chem mean bias and associated spatial standard deviation are inset in the bottom panel.



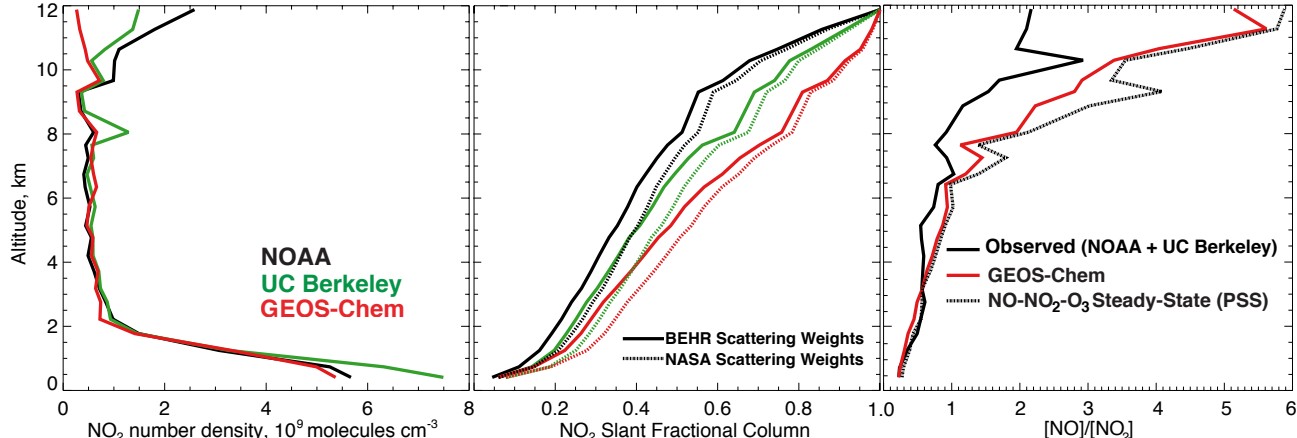

**Figure 6:** Vertical distribution of $NO_2$ over the Southeast US during $SEAC^4RS$ (August-September 2013) and contributions to tropospheric $NO_2$ columns measured from space by OMI. The left panel shows median vertical profiles of $NO_2$ number density measured from the $SEAC^4RS$ aircraft by the NOAA and UC Berkeley instruments and simulated by GEOS-Chem. The middle panel shows the fractional contribution of $NO_2$ below a given altitude to the total tropospheric $NO_2$ slant column measured by OMI, accounting for increasing sensitivity with altitude as determined from the retrieval scattering weights. The right panel shows the mean vertical profiles of the daytime $NO/NO_2$ molar concentration ratio in the aircraft observations (NOAA for NO and UC Berkeley for $NO_2$) and in GEOS-Chem. Also shown is the ratio computed from $NO-NO_2-O_3$ photochemical steady state (PSS) as given by reactions (5)+(7).





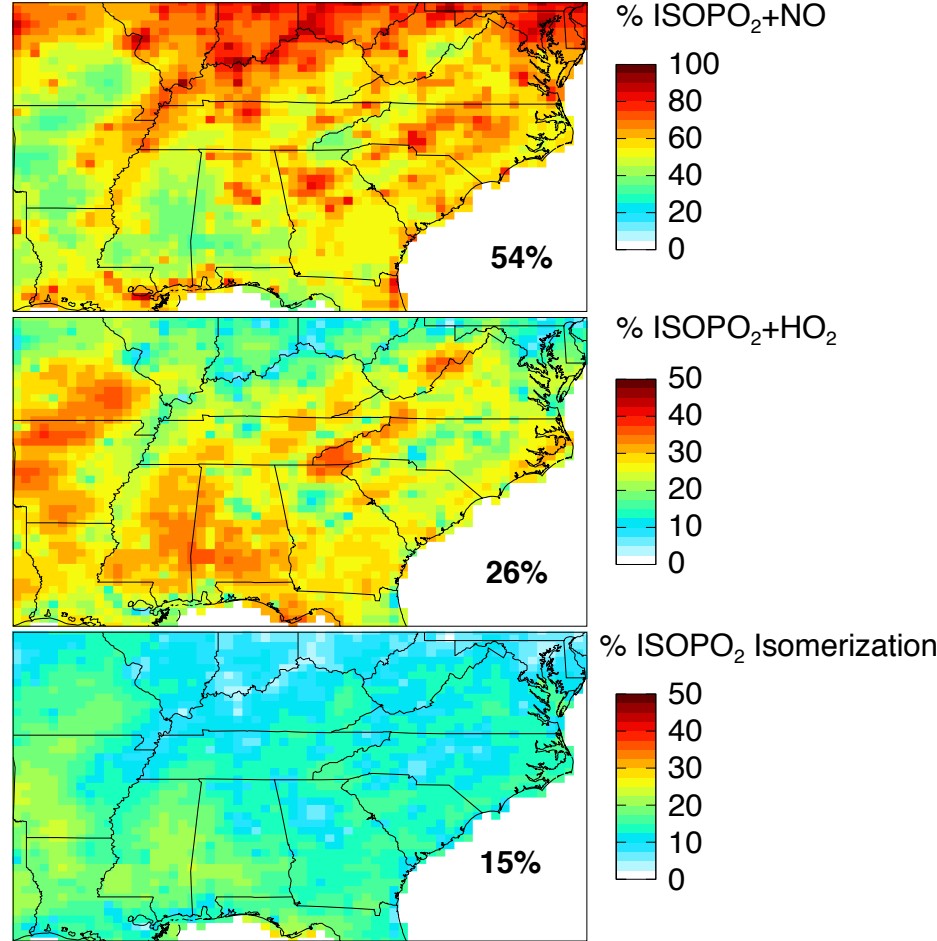

**Figure 7:** Branching ratios for the fate of the isoprene peroxy radical (ISOPO$_2$) as simulated by GEOS-Chem over the Southeast US for August-September 2013. Values are percentages of ISOPO$_2$ that react with NO, HO$_2$, or isomerize from the total mass of isoprene reacting over the domain. Note the difference in scale between the top panel and the lower two panels. Regional mean percentages for the Southeast US are shown inset. They add up to less than 100% because of the small ISOPO$_2$ sink from reaction with other organic peroxy radicals (RO$_2$).





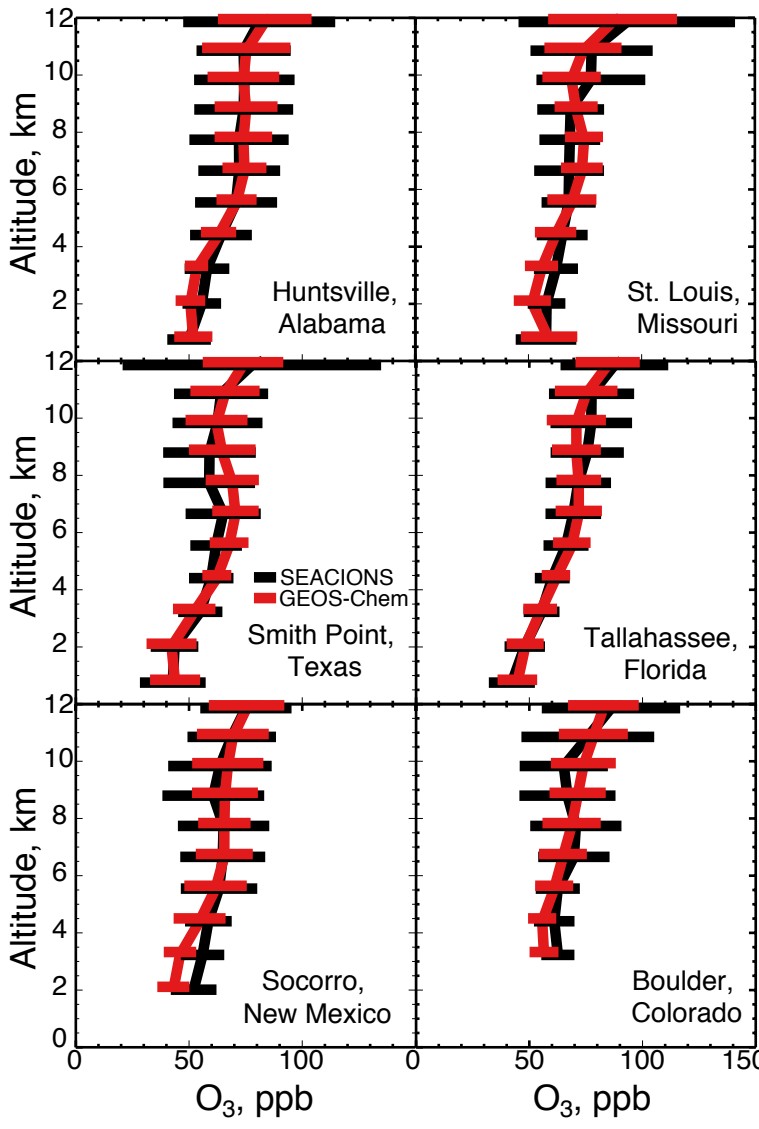

**Figure 8:** Mean ozonesonde vertical profiles at the US SEACIONS sites (http://croc.gsfc.nasa.gov/seacions/) during the SEAC[4]RS campaign in August-September 2013. An average of 25 sondes were launched per site between 11am and 2pm local time. Ozonesondes at Smith Point, Texas were only launched in September. Model values are coincident with the launches. Also shown are standard deviations.





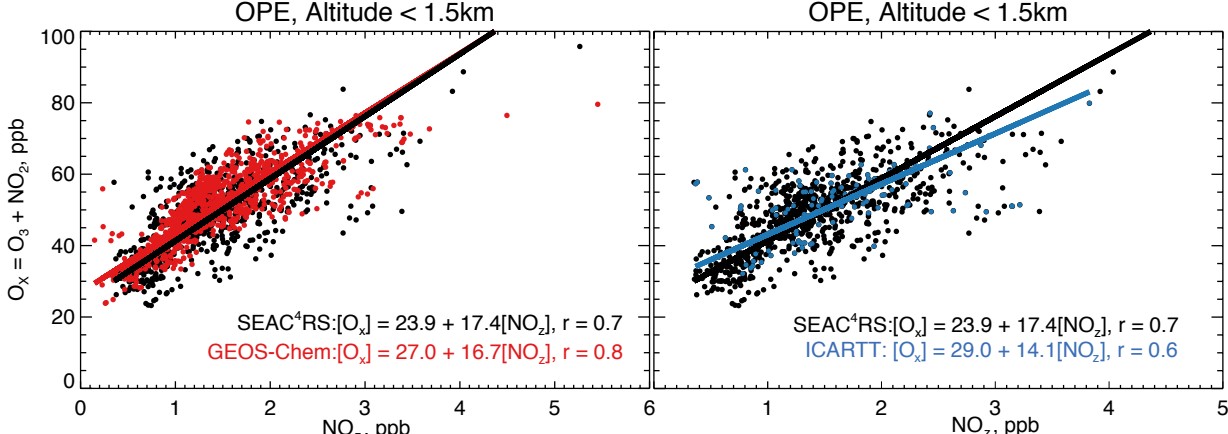

**Figure 9:** Ozone production efficiency (OPE) over the Southeast US in summer estimated from the relationship between odd oxygen ($O_x$) and the sum of $NO_x$ oxidation products ($NO_z$) below 1.5 km altitude. The left panel compares SEAC[4]RS observations to GEOS-Chem values for August-September 2013 (data from Figure 2). The right panel compares SEAC[4]RS observations to INTEX-NA aircraft observations collected over the same Southeast US domain in summer 2004 (Singh et al., 2006). $NO_z$ is defined here as $HNO_3$ + PAN + alklynitrates, all of which were measured from the SEAC[4]RS and INTEX-NA aircraft. The slope and intercept of the reduced-major-axis (RMA) regression are provided inset with the correlation coefficient (r). Observations for INTEX-NA were obtained from ftp://ftp-air.larc.nasa.gov/pub/INTEXA/.

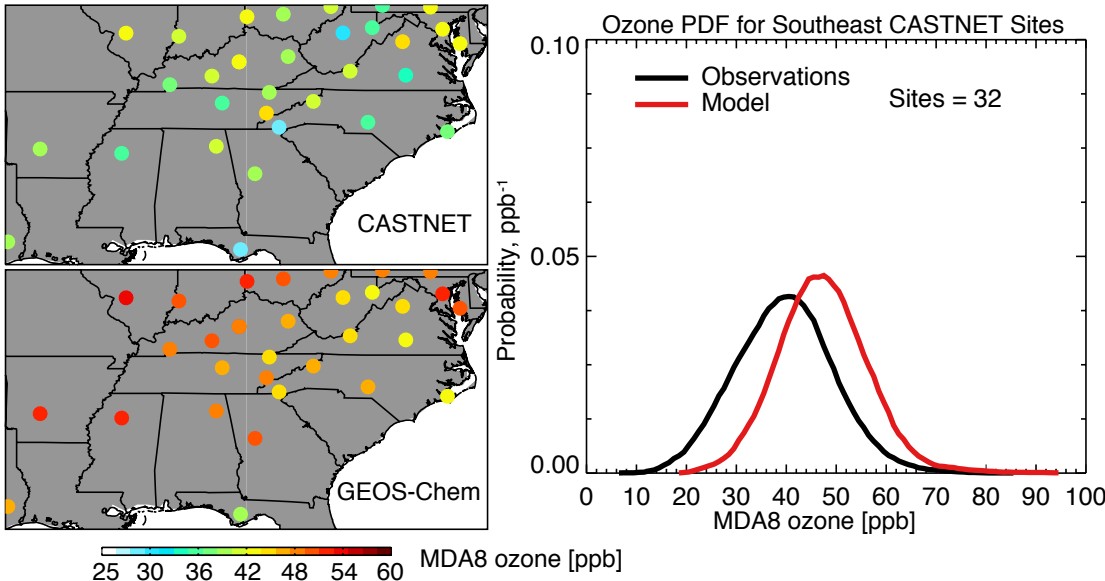

**Figure 10:** Maximum daily 8-h average (MDA8) ozone concentrations at the 32 CASTNET sites in the Southeast US in June-August 2013. The left panels show seasonal mean values in the observations and GEOS-Chem. The right panel shows the probability density functions (pdfs) of daily values at the 32 sites.



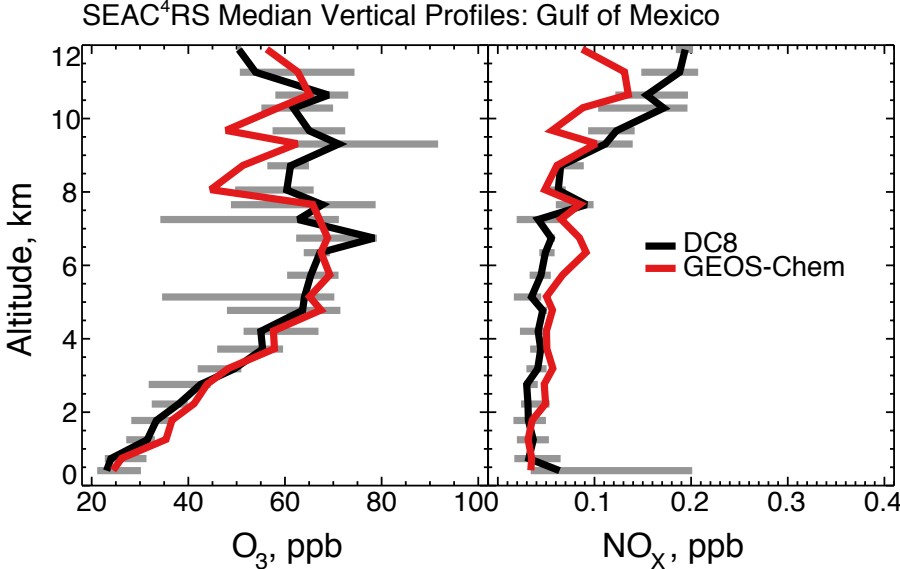

**Figure 11**: Median vertical profiles of ozone and NO$_x$ concentrations over the Gulf of Mexico during SEAC[4]RS. Observations are from four SEAC[4]RS flights over the Gulf of Mexico (August 12, September 4, 13, 16). GEOS-Chem model values are sampled along the flight tracks. The 25[th] and 75[th] percentiles of the aircraft observations are shown as horizontal bars.

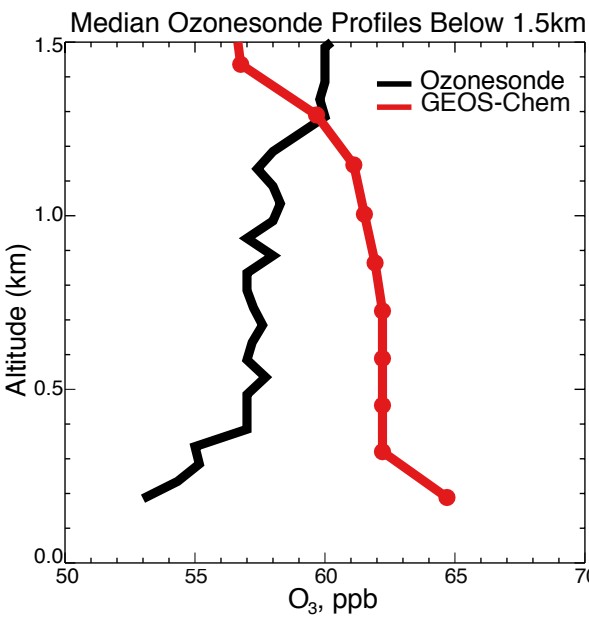

**Figure 12:** Median vertical profile of ozone concentrations over St. Louis, Missouri and Huntsville, Alabama during August and September 2013. Observations from SEACIONS ozonesondes launched between 10am and 1pm local time (n = 57 launches) are compared to GEOS-Chem results sampled at the times of the ozonesonde launches and at the vertical resolution of the model (10 layers below 1.5km, red circles). The ozonesonde data are shown at 50m resolution. Altitude is above local ground level. There are no ozonesonde data below 200.

