# Peer review of "Why do Models Overestimate Surface Ozone in the Southeastern United States?"

_Atmospheric Chemistry and Physics, 2016_

## Referee Comment (RC1) · Anonymous Referee #1 · 12 Apr 2016

The authors use aircraft, surface, satellite and ozonesonde observations to investigate factors controlling surface ozone concentrations in the Southeast US. This is done by comparison to a state-of-the-art chemical transport model. One of the major findings of the study is a high bias of the EPA National Emission Inventory for NOx, most probably due to an overestimation of industrial and mobile sources. Further results are a deviation between NO2 observations and the NO-NPO2-O2 photochemical steady state in the upper troposphere, the role of the partial separation of isoprene and NOx emissions on isoprene chemistry and the effect of NOx reductions on ozone production efficiency. The data analysis is sound and the paper is well written. Thus this manuscript should be published after some minor revisions.

[Figure]

Actually the only criticism that I have is the statement, that ROx chemistry has only a minor role for the NO/NO2 ratio at high altitudes. I agree, that NO2 photolysis and the NO + O3 reaction might be dominant, but due to the low temperatures the later reaction is slower in the UT. Observations indicate that HOx (and most likely RO2) are often enhanced in the UT due to convective injection of precursors. Thus it would be interesting to quantify the role of HO2 and RO2 for the NO/NO2 ratio in the UT.

A minor point is that the titles of chapter 6 and 7 are identical.

---

## Referee Comment (RC2) · Anonymous Referee #3 · 13 Apr 2016

This study utilizes multiple observational data sets and the GEOS-Chem chemical transport model to understand the factors leading to a high modeled ozone bias in the southeastern United States. This is an important problem, as models have had this overestimate for quite some time but it has been difficult to reduce. The paper analyzes model changes to understand and reduce this bias, including (1) changing the National Emission Inventory (NEI) for NOx from the US Environmental Protection Agency (EPA), (2) changing the isoprene chemical mechanism. As a result, this is primarily a model evaluation paper with some sensitivity tests to improve the model bias. However, despite this work, it was still not clear to me how this paper has improved our understanding of the science in the region and how others can learn from these

studies to improve the Southeastern US ozone bias. Overall, this paper still requires some additional work and major revisions are required to make this paper acceptable for ACP.

There are several weaknesses in supporting the assumptions in the manuscript. One main flaw is the changing of the NEI for NOx is not well addressed and justified. The necessity of the large reduction of the NEI for NOx in the Southeast and nationally is the most important conclusion in the manuscript. However, no supporting information is utilized to verify why a reduction of 60% was suggested, besides finding it a close match to observations. The feasibility of implementing the same reduction percentages on all the other sources besides power plant emissions also needs to be justified. In addition, it is unclear how the NEI11 is scaled to the 2013 emission, which is a fundamental piece of information to know before further modifications on the NOx emissions.

One other crucial problem is about the vertical mixing. The authors include this as part of the title, and provide this as a main explanation for the model-measurement bias. Yet it is barely discussed in the manuscript, for example, it is only mentioned briefly in two paragraphs and no discussion what the assimilated vertical mixing from GEOS actually looks like. If the authors think that this is an important factor, then they should discuss what the modeled values are and why they think they are overestimated. Additionally, there is no discussion on how the driving meteorology influences the near-surface turbulent mixing, which is likely important in the reanalysis data they are using. Since a large amount of ozone is produced near the surface, this section will be improved with addressing the effects of both turbulent mixing and surface ozone chemistry to understand the vertical profiles in Figure 12. In addition, the manuscript notes that daytime mixing depths are reduced by 40% in the meteorological setup of the model. It would be helpful to explain how this change influence the dynamics below the boundary layer, which further impact the vertical mixing of ozone.

Before the final publication, I recommend the manuscript to address these key scientific questions and other more minor discussion comments.

[Figure]

1. The study calculates that 54% of the ISOPO2 radical reacts through the high-NOx pathway compared with 62% before the NEI11 adjustment, and states the influence of changing NOx emission on the high-NOx pathway is weak. Since the paper attributes this weak dependence to the spatial segregation between isoprene and NOx emissions, it would be helpful to compare the distribution of segregation with the high-NOx pathway results to confirm this conclusion.

2. The bias between simulations and observations is still large in some regions, as shown in both Figure 3 and Figure 4. In Figure 4, the manuscript uses an uneven color bar for NOx, making it hard to distinguish the differences between observed and simulated NOx. It would be clear to identify those differences using constant color bar scale for NOx or provide more color contours, or to make a contour plot for the differences of NOx and O3 between simulations and observations. With the biases in Figure 3 and 4, the changes on NOx emissions could have a regional dependence.

3. The sentence "no indication of regional patterns of model bias that would point to the need for a more selective adjustment of NOx emissions" is not clear to me. It would be better to draw a conclusion about regional patterns after analyzing the model biases in Figure 3 and 4.

4. The domains of the maps are not consistent in Figures 1, 3, 4, 5 and 7, which is confusing as they come from the same simulations. Also, it would be helpful to add lat/lon labels for the contour maps.

Technical corrections:

1. Same title for Section 6 and Section 7. Please clarify the differences between these two sections.

---

## Author Comment (AC1) · 23 Jun 2016

We thank the two reviewers for their careful reading of the manuscript and their detailed comments. Our responses are shown below in blue, with new text in bold.

**Anonymous Referee #1**

Actually the only criticism that I have is the statement, that $RO_x$ chemistry has only a minor role for the $NO/NO_2$ ratio at high altitudes. I agree, that $NO_2$ photolysis and the $NO + O_3$ reaction might be dominant, but due to the low temperatures the later reaction is slower in the UT. Observations indicate that $HO_x$ (and most likely $RO_2$) are often enhanced in the UT due to convective injection of precursors. Thus it would be interesting to quantify the role of $HO_2$ and $RO_2$ for the $NO/NO_2$ ratio in the UT.

We have revised Section 4 to include a discussion of $HO_2$ and $RO_2$. We have added the following paragraph, with the revisions to Figure 6 below.

**"Zhu et al. (2016) found that GEOS-Chem underestimates the observed HCHO concentrations in the upper troposphere during SEAC4RS by a factor of 3, implying that the model underestimates the HOx source from convective injection of HCHO and peroxides (Prather and Jacob, 1997; Müller and Brasseur, 1999). $HO_2$ observations over the central US in summer during the SUCCESS aircraft campaign suggest that this convective injection increases $HO_x$ concentrations in the upper troposphere by a factor of 2 (Jaeglé et al., 1998). The bottom right panel of Figure 6 shows median modeled and observed vertical profiles of the $HO_x$ reservoir hydrogen peroxide ($H_2O_2$) during $SEAC^4RS$ over the Southeast US. GEOS-Chem underestimates observed $H_2O_2$ by a mean factor of 1.7 above 8km. The middle right panel of Figure 6 shows the predicted $[NO]/[NO_2]$ ratio if modeled convective injection of $HO_2$ and $RO_2$ precursors is underestimated by a factor of 2. While such an underestimate is insufficient to reconcile simulated and observed $[NO]/[NO_2]$ concentration ratios, the contribution to the $[NO]/[NO_2]$ ratio from Reaction 6 would be much more significant than previously estimated."**

[Figure]

**Figure 1: Vertical distribution of NO$_2$ over the Southeast US during SEAC$^4$RS (August-September 2013) and contributions to tropospheric NO$_2$ columns measured from space by OMI. The top left panel shows median vertical profiles of NO$_2$ number density measured from the SEAC$^4$RS aircraft by the NOAA and UC Berkeley instruments and simulated by GEOS-Chem. The top right panel shows the fractional contribution of NO$_2$ below a given altitude to the total tropospheric NO$_2$ slant column measured by OMI, accounting for increasing sensitivity with altitude as determined from the retrieval scattering weights. The bottom left panel shows the median vertical profiles of the daytime [NO]/[NO$_2$] molar concentration ratio in the aircraft observations (NOAA for NO and UC Berkeley for NO$_2$) and in GEOS-Chem. Also shown is the ratio computed from NO-NO$_2$-O$_3$ photochemical steady state (PSS) as given by reactions (5)+(7) (blue) and including reaction (6) with doubled HO$_2$ and RO$_2$ concentrations above 8km (purple). The bottom right panel shows the median H$_2$O$_2$ profile from the model and from the SEAC4RS flights over the Southeast US. H$_2$O$_2$ was measured by the Caltech CIMS (see Figure 2).**

A minor point is that the titles of chapter 6 and 7 are identical.

This typo has been fixed. Section 7 now reads **"7. Implications for ozone: surface air"**

**Anonymous Referee #3**

One main flaw is the changing of the NEI for NOx is not well addressed and justified. The necessity of the large reduction of the NEI for NOx in the Southeast and nationally is the most important conclusion in the manuscript. However, no supporting information is utilized to verify why a reduction of 60% was suggested, besides finding it a close match to observations. The feasibility of implementing the same reduction percentages on all the other sources besides power plant emissions also needs to be justified. In addition, it is unclear how the NEI11 is

scaled to the 2013 emission, which is a fundamental piece of information to know before further modifications on the NOx emissions.

We focus in this paper on the Southeast U.S., where emissions of non-anthropogenic $NO_x$ are small compared to NEI11v1 emissions. We state on line 26 on page 5 the following: **"Errors in $NO_x$ sources from soils, wildfire, or lightning cannot account for the overestimate because their magnitudes are small relative to fuel combustion, as shown below."**

We have added a clarifying sentence for the 2013 emission factor on Page 5 line 17: **"The scaling factor for 2013 for $NO_x$ emissions is 0.89. Further information on the use of the NEI11v1 in GEOS-Chem can be found here: http://wiki.seas.harvard.edu/geos-chem/index.php/EPA/NEI11_North_American_emissions/."**

One other crucial problem is about the vertical mixing. The authors include this as part of the title, and provide this as a main explanation for the model-measurement bias. Yet it is barely discussed in the manuscript, for example, it is only mentioned briefly in two paragraphs and no discussion what the assimilated vertical mixing from GEOS actually looks like. If the authors think that this is an important factor, then they should discuss what the modeled values are and why they think they are overestimated. Additionally, there is no discussion on how the driving meteorology influences the near-surface turbulent mixing, which is likely important in the reanalysis data they are using. Since a large amount of ozone is produced near the surface, this section will be improved with addressing the effects of both turbulent mixing and surface ozone chemistry to under- stand the vertical profiles in Figure 12. In addition, the manuscript notes that daytime mixing depths are reduced by 40% in the meteorological setup of the model. It would be helpful to explain how this change influence the dynamics below the boundary layer, which further impact the vertical mixing of ozone.

We agree with the comment about the title and have changed the paper title to the following: **Why do Models Overestimate Surface Ozone in the Southeastern United States?**

We have revised Page 3 lines 18-21 to include a more detailed description of model vertical mixing and the reduction in boundary layer height.
**"Turbulent boundary layer mixing follows a non-local parameterization based on K-theory (Holtslag and Boville, 1993) implemented in GEOS-Chem by Lin and McElroy (2010). Daytime mixing depths are reduced by 40% as described by Kim et al. (2015) and Zhu et al. (2016) to match lidar observations of boundary layer height from SEAC[4]RS."**

We have changed to discussion of vertical mixing throughout the paper to include a discussion of both excessive vertical mixing and chemical production of ozone likely due to excessive dryness in the model. We replace line 7 on page 2 with the following: **"This may be caused by excessively dry conditions in the model, representing another factor important in the simulation of surface ozone."**
We replaced the paragraph on Page 12 line 19 with the following:
**"It appears instead that there is a model bias in boundary layer vertical mixing and chemical production. Figure 12 shows the median ozonesonde profile at a higher vertical resolution over the Southeast US (Huntsville, Alabama and St. Louis, Missouri sites)**

**during SEAC[4]RS as compared to GEOS-Chem below 1.5 km. The ozonesondes indicate a decrease of 7 ppb from 1.5 km to the surface, whereas GEOS-Chem features a reverse gradient of increasing ozone from 1.5 to 1 km with flat concentrations below. Preliminary inspection suggests that this may reflect excessively dry conditions in the GEOS-5.11.0 meteorological fields, promoting boundary layer production and vertical mixing of ozone. Such a bias might not be detected in the aircraft data, collected mainly under fair weather conditions."**

We replace the concluding sentences on Page 13 line 33 with the following:
**"This may be due to excessively dry conditions in the GEOS meteorological data used to drive GEOS-Chem, resulting in excessive boundary layer ozone production and mixing. Such a bias may not be detected in the aircraft data, generally collected under fair-weather conditions. Investigating this source of bias and its prevalence across models will be the topic of a follow-up paper."**

The study calculates that 54% of the ISOPO2 radical reacts through the high-NOx pathway compared with 62% before the NEI11 adjustment, and states the influence of changing NOx emission on the high-NOx pathway is weak. Since the paper attributes this weak dependence to the spatial segregation between isoprene and NOx emissions, it would be helpful to compare the distribution of segregation with the high-NOx pathway results to confirm this conclusion.

This work is done in Yu et al, 2016. We have clarified this sentence on page 10 line 28: **It now reads, "The lack of dominance of the high-NO$_x$ pathway is due in part to the spatial segregation of isoprene and NO$_x$ emissions (Yu et al., 2016)."**

The bias between simulations and observations is still large in some regions, as shown in both Figure 3 and Figure 4. In Figure 4, the manuscript uses an uneven color bar for NOx, making it hard to distinguish the differences between observed and simulated NOx. It would be clear to identify those differences using constant color bar scale for NOx or provide more color contours, or to make a contour plot for the differences of NOx and O$_3$ between simulations and observations. With the biases in Figure 3 and 4, the changes on NOx emissions could have a regional dependence.

We have changed the scale on Figure 4 for NO$_x$ to a linear rather than a log scale.

We provide the spatial correlation coefficient for the Southeast US and the continental US inset in Figure 3 to show the good agreement between model and observation.

We state on Page 7 line 21 that for NO$_x$, "The spatial correlation coefficient is 0.71." I have added the following to the caption of Figure 4 to clarify the good spatial agreement between model and observation for both NO$_x$ and O$_3$: **"The spatial correlation coefficient is 0.71 for both NO$_x$ and O$_3$. The normalized mean bias is -11.5% for NOx and 4.5% for O$_3$."**

We add the following sentence to Page 6 line 1: **"There is no information in the spatial pattern of bias that would warrant a more location-specific or source-specific reduction."**

The sentence "no indication of regional patterns of model bias that would point to the need for a more selective adjustment of NOx emissions" is not clear to me. It would be better to draw a conclusion about regional patterns after analyzing the model biases in Figure 3 and 4.

See reply to #4 above.

We added the following clarification on page 7 line 13: **"We see from Figure 3 that the model with decreased NO$_x$ emissions reproduces the spatial variability in the observations with minimal bias over the Southeast US domain shown in Figure 1 and across the rest of the country."**

The domains of the maps are not consistent in Figures 1, 3, 4, 5 and 7, which is confusing as they come from the same simulations. Also, it would be helpful to add lat/lon labels for the contour maps.

The maps are intended to either show the CONUS, Southeast US, and/or or the Gulf of Mexico. We have added lat/lon labels to the contour maps for Figures 1, 4, 5, and 7 to clarify their domains.

Technical corrections:

1. Same title for Section 6 and Section 7. Please clarify the differences between these two sections.
   a. **"7. Implications for ozone: surface air"**

---

## Editor Decision (ED1)

*Second review on "*Why do Models Overestimate Surface Ozone in the Southeastern United States?*" by Travis et al.*

The manuscript has been revised according to most of the comments. The authors' efforts on the revisions are appreciated. However, the first question in the previous review is still not answered. In addition, the title has been changed trying to clear the focus of this manuscript. But no solid scientific explanation is actually provided to answer the question in the new title. I recommend the manuscript to clarify these two problems before the final publication.

1. About the first question from the previous review, still, no supporting information is utilized to verify why a reduction of 60% on both the mobile and industry emissions was suggested. In the Supplement, the authors replied to this comment by "We focus in this paper on the Southeast U.S., where emissions of non-anthropogenic NOx are small compared to NEI11v1 emissions". But no citation or other information is provided to support this argument. The authors also added a statement "Errors in NOx sources from soils, wildfire, or lightning cannot account for the overestimate because their magnitudes are small relative to fuel combustion". The total contribution of soil, fire and fertilizer in Figure 1 is up to 32% after reducing the emissions from mobile and industry by 60%. Even before the reduction, they contribute about 19%, which is not trivial. The second paragraph in section 2.3 provides some preliminary review on the emission sources. A more convincing

literature review on the changes and evaluations of different emissions is
necessary and is expected to answer this question.

2.  According to the new title, the manuscript is aim to explain the modeling bias
    in surface ozone. Assuming the 60% reduction in the emission inventory is
    well justified, there are still discrepancies in simulating the ozone profiles
    below 1.5 km and the distribution of ozone in the Southeast US.

    Based on "preliminary inspection", the authors proposed the near-surface
    ozone bias "may be due to excessively dry conditions in the GEOS
    meteorological data used to drive GEOS-Chem, resulting in excessive
    boundary layer ozone production and mixing". The terms "preliminary
    inspection" and "excessively dry conditions" are unclear. A figure comparing
    water vapor profiles from the GEOS meteorological data and the observation
    would be helpful to verify the dry conditions. As this conclusion is included in
    the abstract, explanations on how this dry condition leads to excessive ozone
    production and mixing are expected. In addition, the sentence "such a bias
    might not be detected in the aircraft data" does not make sense. The aircraft
    detects the real atmospheric environment, not the bias.

    About spatial distribution, the comparisons in Figure 3 and Figure 4 show
    obvious differences in some regions, e.g., in Georgia State in Figure 4 (up to
    about 50% bias in NO and 20% in $O_3$), indicating a location-specific reduction

could be required. The term "minimal bias" is not appropriate. These differences should be quantified and explanations should be provided.

**Technical corrections**

- Line 26 "The resulting US anthropogenic NOx emissions from fuel combustion for 2013 total 1.7 Tg N a$^{-1}$" is not a full sentence.

---

## Author Response (AR2)

**Dr. Ganzeveld,**

**Thank you for your comments. Our responses are in blue, with new text in bold.**

Dear author, co-authors, I have read the reviews of your ms as well as your response to the comments provided by both reviewers and your revised version of the ms. One of the reviewers was apparently appreciating the contents of your paper and only had some minor comment which you addressed well in your response and revision of the ms. The other reviewer had some major criticism on especially the explanation/justification of issues on the emissions and role of vertical mixing in explaining some of the model biases in surface ozone in the SE USA. In response to these comments you bring up an interesting reasoning that to some extent might indeed explain why models such as GEOS-CHEM overestimates surface ozone in this region of the US. However, it also raises a number of additional questions that should be addressed; You are indicating that one of the main reasons for the overestimation of surface (or boundary layer) ozone in GEOS-CHEM is the apparent too dry input meteorology. This would according to your explanation result in a too strong chemical production of ozone as well as too strong boundary layer mixing. How would you diagnose that?

We have modified the conclusion on Page 14 line 1 to read the following:

**A comparison of GEOS-5.11.0 with observations of soil moisture, surface temperature, and other relevant meteorological variables to determine the source of bias and its prevalence across models will be the topic of a follow-up paper.**

You indicate that the boundary layer mixing has been modified to result in the simulation of a shallower BL depth in agreement with the observations but assessing boundary layer mixing requires to also assess vertical profiles in potential temperature, moisture and concentrations of tracers such as ozone. The latter explanation on the too dry conditions and impact on BL mixing then introduces some controversy since with a too strong boundary layer mixing you would detrain/export some of the ozone precursors much more efficiently out of the boundary layer (as well as the excessive ozone if this would be indeed higher in the BL compared to free tropospheric ozone levels). But also now that you bring in

this explanation for the overestimation of surface ozone, it triggers a major new question: how does the too dry meteorology in GEOS-CHEM affect atmosphere-biosphere exchange processes. The too dry (and warm?) conditions might strongly enhance some of the BVOC and soil-biogenic N emissions, although you indicate that the latter term is probably of minor importance compared to the anthropogenic emissions. It is indicated in the paper that some of the BVOC emissions have been scaled down, a correction needed to deal with the bias in the input meteorology?

The scaling is based on observations and is only attributed to uncertainty in the MEGAN model at this time. We have modified the explanation of BVOC emissions on Page 6 line 24 as follows: We reduce MEGAN v2.1 isoprene emissions by 15% to better match SEAC[4]RS observations **of isoprene fluxes from the Ozarks** (Wolfe et al., 2015) **and observed formaldehyde** (Zhu et al., 2016).

The other issue that requires than further analysis and discussion; how is the dry deposition is affected by these too dry meteorological conditions in the model? You indicate that GEOS-CHEM uses an implementation of the Wesely scheme and where you have indicated that there have been some modifications to properly simulated the observed dry deposition for one specific site in the US. How much is the overall dry deposition for the whole region changed by these modification and how much does this affect surface ozone levels compared to some of the other scalings, e.g., in NOx emissions and boundary layer depth?

On Page 16, line 13 we have added the following sentence:
**The improvements to dry deposition described in Section 2.2 minimally reduce (approximately 1 ppb) GEOS-Chem ozone compared to SEAC[4]RS boundary layer and CASTNET surface MDA8 ozone observations. The reduction of daytime mixing depths described in Section 2 results in a small increase in mean MDA8 ozone (approximately 2 ppb) due to an increase in ozone production at the surface.**

It is interesting to note that you would actually expect that, due to the too dry conditions you would

underestimate dry deposition when you would consider the potential impact of soil moisture and other hydrological drivers of vegetation uptake such as vapor pressure deficit on dry deposition but I assume this is not included in GEOS-CHEM.

Correct.

One of your main conclusions is that the EPA NOx emission inventory seems to give a too large source of NOx for the US. Given the potential implications of this finding I appreciate the comments by the second reviewer to secure a further support of your case that it is indeed the emissions that are overestimated and not a selection of processes that are not properly represented in your modelling system. I do though also see that this could easily result in a couple of further follow-up studies, as you also indicate in the conclusions. Including in your discussions some more explicit description on what would be needed to further corroborate your findings would make a further valuable modification of this ms.

We have added the following sentence to the concluding paragraph: **Further studies should evaluate the EPA NO$_x$ inventory for other years and seasons and explore potential reasons for additional inventory or model bias.**

My specific feedback is that I deem being essential is to conduct some detailed analysis for some specific locations (with field observations with chemistry and micro-BL meteorology observations) to assess the diurnal cycle in dry deposition (not only of ozone but also precursors and products), boundary layer evolution (not only BL depth but also vertical profiles) and analysis of the diurnal cycles in the ozone process tendencies.

We have added the following sentence to Page 5 line 10: **The diurnal cycle of dry deposition in GEOS-Chem compares well with the observations from SOAS (Nguyen et al., 2015).** Analysis of boundary layer evolution and ozone process tendencies will be part of the follow-up study.

I anyhow intend to invite the second reviewer to provide an evaluation of the revised version of the ms for making my final decision but before doing so, want to raise these points that you can potentially include in a further updated revision/response.

We have uploaded revisions to the manuscript.

[revised manuscript text omitted]

---

## Author Response (AR3)

**Dear Reviewer,**

**Thank you for your comments. Our responses are in blue, with new text in bold. The line numbers and page references listed refer to the marked up manuscript following this reply.**

The manuscript has been revised according to most of the comments. The authors' efforts on the revisions are appreciated. However, the first question in the previous review is still not answered. In addition, the title has been changed trying to clear the focus of this manuscript. But no solid scientific explanation is actually provided to answer the question in the new title. I recommend the manuscript to clarify these two problems before the final publication.

1. About the first question from the previous review, still, no supporting information is utilized to verify why a reduction of 60% on both the mobile and industry emissions was suggested. In the Supplement, the authors replied to this comment by "We focus in this paper on the Southeast U.S., where emissions of non-anthropogenic NOx are small compared to NEI11v1 emissions". But no citation or other information is provided to support this argument. The authors also added a statement "Errors in NOx sources from soils, wildfire, or lightning cannot account for the overestimate because their magnitudes are small relative to fuel combustion". The total contribution of soil, fire and fertilizer in Figure 1 is up to 32% after reducing the emissions from mobile and industry by 60%. Even before the reduction, they contribute about 19%, which is not trivial. The second paragraph in section 2.3 provides some preliminary review on the emission sources. A more convincing literature review on the changes and evaluations of different emissions is necessary and is expected to answer this question.

The second paragraph in section 2.3 cites the relevant field studies known to us at this time on the apparent overestimate in the NEI $NO_x$ emissions inventory. We have changed the sentence on Page 13 line 26 to the following to clarify our meaning.

**"Several local studies in recent years have found that NEI NO$_x$ emissions for mobile sources may be too high by a factor of two or more (Castellanos et al, 2011; Fujita et al., 2012; Brioude et al., 2013; Anderson et al., 2014)."**

We have added the following discussion to discuss the uncertainty in the NO$_x$ scaling factor for NEI11v1 due to the presence of soil NO$_x$. We now include a scenario in which soil NO$_x$ is underestimated by 100%, and give the required NEI scale factor to achieve the same NO$_x$ simulation.

*We revise our conclusion in the abstract.*

Page 7 line 8, **"Our results indicate that NEI NO$_x$ emissions from mobile and industrial sources must be reduced by 30-60%, the range reflecting uncertainties in soil NO$_x$ emissions."**

*We revise our discussion of emissions and our literature review.*

Page 13 line 20, **"Initial implementation of the above inventory in GEOS-Chem resulted in a 60-**

**70% overestimate of NO$_x$ and HNO$_3$ measured from the SEAC[4]RS DC-8 aircraft, and a 70% overestimate of nitrate (NO$^{3-}$) wet deposition fluxes measured by the National Acid Deposition Program (NADP) across the Southeast US. Correcting this bias required a ~40% decrease in surface NO$_x$ emissions. Soil and fertilizer NO$_x$ emissions (18% of total NO$_x$ emissions in the Southeast) and open fires (2%) would be insufficient to correct this bias. Emissions from power**

**plant stacks are directly measured but account for only 12% of NEI NO$_x$ emissions on an annual basis (EPA, 2015). Several local studies in recent years have found that NEI NO$_x$ emissions for mobile sources may be too high by a factor of two or more (Castellanos et al, 2011; Fujita et al., 2012; Brioude et al., 2013; Anderson et al., 2014). We can achieve the required 40% decrease in total NO$_x$ emissions by reducing NEI emissions from mobile and industrial sources (all sources**

**except power plants) by 60%, or alternatively by reducing these sources by 30% and zeroing out soil and fertilizer NO$_x$ emissions. Here we choose to do the former. There is enough spatial overlap between anthropogenic and soil emissions that we cannot readily arbitrate between these two scenarios. Comparisons with observations will be presented in the next Section."**

*We revise our discussion of our results given a potential overestimate in soil nox.*

Page 15, line 13, **"Results are very similar if we decrease the non-power plant NEI fuel emissions by only 30% and zero out soil and fertilizer emissions. Thus the required decrease of $NO_x$ emissions may involve an overestimate of both anthropogenic and soil emissions."**

Page 16, line 8, **"There are no obvious spatial patterns of model bias that would point to specific source sectors as responsible for the $NO_x$ emission overestimate, beyond the blanket 30-60% decrease of non-power plant NEI emissions needed to correct the regional emission total."**

*We revise our conclusion.*

Page 24, line 8, **"Presuming no error in emissions from large power plants with continuous emission monitors (14% of unadjusted NEI inventory), we find that emissions from other industrial sources and mobile sources must be 30-60% lower than NEI values, where the range reflects uncertainty in the contribution from soil $NO_x$ emissions. We thus estimate that anthropogenic fuel $NO_x$ emissions in the US in 2013 were 1.7-2.6 Tg N $a^{-1}$, as compared to 3.5 Tg N $a^{-1}$ given in the NEI."**

2. According to the new title, the manuscript is aim to explain the modeling bias in surface ozone. Assuming the 60% reduction in the emission inventory is well justified, there are still discrepancies in simulating the ozone profiles below 1.5 km and the distribution of ozone in the Southeast US.

Based on "preliminary inspection", the authors proposed the near-surface ozone bias "may be due to excessively dry conditions in the GEOS meteorological data used to drive GEOS-Chem, resulting in excessive boundary layer ozone production and mixing". The terms "preliminary inspection" and "excessively dry conditions" are unclear. A figure comparing water vapor profiles from the GEOS meteorological data and the observation would be helpful to verify the dry conditions. As this conclusion is included in the abstract, explanations on how this dry condition leads to excessive ozone production and mixing are expected. In addition, the sentence "such a bias might not be detected in the aircraft data"

does not make sense. The aircraft detects the real atmospheric environment, not the bias.

We have removed specific speculation on the ozonesonde/model discrepancies for discussion in a follow-up paper. The goal of this paper is to discuss the many reasons speculated for the lack of agreement between models and measured surface ozone in the Southeast US and make progress on this important issue. We have changed the following discussion of dry conditions throughout the paper to only discuss the clear evidence of excessive mixing and excessive net ozone production in the boundary layer. We now avoid speculation of potential dry conditions and discussion of the meteorological fields in GEOS-Chem.

*We revise our conclusion in the abstract.*
Page 7 line 18, **"This bias may reflect a combination of excessive vertical mixing and net ozone production in the model boundary layer."**

*We revise our discussion of Figure 12.*
Page 23 line 21, **"This implies a combination of two model errors in the boundary layer: (1) excessive vertical mixing, (2) net ozone production whereas observations indicate a net loss."**

*We revise our conclusion.*
Page 25, line 15, **"This may be due to excessive boundary layer mixing and net ozone production in the model. Excessive mixing in GEOS-Chem may be indicative of an overestimate of sensible heat flux (Holtslag and Boville, 1993), and thus an investigation of boundary layer meteorological variables is warranted. Such a bias may not be detected in the comparison of GEOS-Chem with aircraft data, generally collected under fair-weather conditions and with minimal sampling in the lower part of the boundary layer. An investigation of relevant meteorological variables and boundary layer source and sink terms in the ozone budget to determine the source of bias and its prevalence across models will be the topic of a follow-up paper. "**

About spatial distribution, the comparisons in Figure 3 and Figure 4 show obvious differences in some regions, e.g., in Georgia State in Figure 4 (up to about 50% bias in NO and 20% in O3), indicating a location-specific reduction could be required. The term "minimal bias" is not appropriate. These differences should be quantified and explanations should be provided.

It is beyond the scope of this work to determine why there are gridbox-specific discrepancies since they are likely due to a variety of factors including model representation errors and day-to-day variation in emission sources and meteorology.

*We have removed the term "minimal bias" from our discussion and replaced it with specific bias values.* Page 15, line 26, **"
[revised manuscript text omitted]

---

## Author Response (AR4)

**Dear Dr. Ganzeveld,**

**Thank you for your comments. Our responses are in blue. The line numbers and page references listed refer to the marked up manuscript.**

5  In your response you indicate that you have included now as an extra analysis one that assumes zero soil NOx emissions. This is clearly a drastic sensitivity analysis where which you justify referring to the large uncertainty in soil NOx emissions. I suggest to change some of the statements about this extra feature of your analysis:

10  Discussion: "Our results indicate that NEI NOx emissions from mobile and industrial sources must be reduced by 30-60%, the range reflecting uncertainties in soil NOx emissions"

Change to: "Our results indicate that NEI NOx emissions from mobile and industrial sources must be reduced by 30-60%, dependent on the assumptions on the contribution by soil NOx emissions"

We have made this change.

Page 13, line 20: "Soil and fertilizer NOx emissions (18% of total NOx emissions in the Southeast) and open fires (2%) would be insufficient to correct this bias."

Change to: "Assuming strongly reduced soil and fertilizer NOx emissions (18% of total NOx emissions in the Southeast) and open fires (2%) also considering the large uncertainty in these emissions would be insufficient to correct this bias.

25  We have made this change.

Line 25: "We can achieve the required 40% decrease in total NOx emissions by reducing NEI emissions from mobile and industrial sources (all sources except power plants) by 60%, or alternatively by reducing

these sources by 30% and zeroing out soil and fertilizer NOx emissions. Here we choose to do the former. There is enough spatial overlap between anthropogenic and soil emissions that we cannot readily arbitrate between these two scenarios."

5  Change to: "We can achieve the required 40% decrease in total NOx emissions by reducing NEI emissions from mobile and industrial sources (all sources except power plants) by 60%, or alternatively by reducing these sources by 30% and zeroing out soil and fertilizer NOx emissions. Since it is apparent that there is some minimum contribution by soil NOx emissions we assessed the impact of the approach reducing the NEI emissions by 60%. The spatial overlap between anthropogenic and soil NOx emissions is such that

10  we cannot readily arbitrate between these two scenarios"

We have made this change.

Conclusions: ", we find that emissions from other industrial sources and mobile sources must be 30-60%

15  lower than NEI values, where the range reflects uncertainty in the contribution from soil NOx emissions."

Change to: ", we find that emissions from other industrial sources and mobile sources must be 30-60% lower than NEI values, dependent on the assumptions on the contribution by soil NOx emissions."

20  We have made this change.

Finally, I see that you have removed the statements about the "excessively dry conditions in GEOS-CHEM". This was also based on the comments raised by the reviewer which was hinting towards a more detailed analysis then of some of the BL meteorological properties to further assess this. However, reading

25  again the revised paper, I would actually suggest you to keep this indication about a potential explanation of some the issues on the simulations of too high ozone in the modelling system since it nicely indicates what should be focus of future research. I would suggest to clearly state that (something like) "It is beyond the scope of the presented study to further analyze this in more detail. However, also recognizing the role

of the hydrological cycle in surface and boundary layer exchange processes such as soil NOx emissions, dry and wet deposition and turbulent transport, this feature should be focus of follow-up studies regarding assessments of air quality over the US using GEOS-CHEM"

5   I am presently working on this problem and am not comfortable making any statement that would turn out to be incorrect. I am also not comfortable attributing the problem as specific to GEOS-Chem since it may be more general than that. We would prefer to conclude with the more general statement, on page 21 line, 6 - **"
[revised manuscript text omitted]